# Support Vector Machines : A more certain estimate of uncertainty

## Abstract

This paper explores the potential of Support Vector Machines (SVMs) from the lens of uncertainty quantification (UQ), for regression and forecasting tasks. Unlike Neural Network (NN) and Random Forest (RF) based approaches, which are commonly used for UQ estimation, SVM-based methods are more stable and sparse, and offer well-defined optimal solutions. This makes SVM models often exhibit much lower model uncertainty than alternative learning models. However, there is only limited literature addressing UQ in SVM-based prediction, particularly in the regression setting. We first provide a comprehensive summary of existing Prediction Interval (PI) estimation and probabilistic forecasting methods developed within the SVM framework. Although SVMs offer globally optimal and stable solutions, the existing literature on UQ within the SVM framework still exhibits several critical gaps. In this work, we also address these gaps and extend contemporary UQ techniques to SVMs, to promote their applicability across diverse domains for more reliable estimation. Our major contributions include the development of sparse SVM models for PI estimation and probabilistic forecasting, an investigation of the role of feature selection in PI estimation, and the extension of SVM regression to the Conformal Regression (CR) setting to construct more stable prediction sets with finite-sample guarantees. Extensive numerical experiments indicate that SVM-based models are a promising choice for UQ modeling in small- and moderate-scale regression tasks, often achieving high-quality PI estimates while maintaining relatively low model uncertainty.

## 1 Introduction

Given the training set $T = \{(x_i, y_i) : x_i \in \mathbf{R}^n, y_i \in \mathbf{R}, i = 1, 2, \ldots, m\}$, sampled independently from the joint distribution of the random variables $(X, Y)$, the goal of the regression task is to estimate a function that predicts the target variable $y$ well based on the input variable $x$. However, in most applications, the prediction of a regression model may not be perfectly accurate due to the random relationship between $Y$ and $X$. For example, predicting the impact of a specific drug on a patient's heart rate based on their Body Mass Index (BMI) may not be accurate and could involve a significant degree of uncertainty. In such cases, quantifying these uncertainties is crucial for making effective decisions.

Prediction Interval (PI) estimation is the main Uncertainty Quantification (UQ) technique in regression tasks. Given a high confidence $1 - \alpha \in (0, 1)$ and a training set $T$, the PI tube is defined as a pair of functions $(f_1, f_2)$. It is said to be well calibrated if it satisfies $P(f_1(X) \leq Y \leq f_2(X)|X) \geq 1 - \alpha$. The objective of PI models is to obtain a PI tube with the minimum possible width while ensuring the target calibration. Therefore, the performance of a PI estimation method is mainly evaluated using two criteria: the Prediction Interval Coverage Probability (PICP), which computes the fraction of $y$ values within the PI tube, and the Mean Prediction Interval Width (MPIW), which quantifies the width of the PI tube.

PI estimation models explore the different characteristics of the conditional distribution $Y|X$, rather than focusing only on $E(Y|X)$, as done in standard regression tasks. The basic approach of PI models involves estimating a pair of quantile functions (Koenker & Bassett Jr, 1978), say $(f_q(x), f_{1+q-\alpha}(x))$, of the conditional distribution $Y|X$, for some $0 \leq q \leq \alpha$, where the $q^{\text{th}}$ quantile function for a given $x$ is $f_q(x)$, defined as the

infimum over all functions satisfying $P(y \leq f(x)|x) = q$. For time-series data, estimating the PI for future observations using an auto-regressive approach is referred to as probabilistic forecasting.

A well-calibrated High-Quality (HQ) PI guarantees the target coverage level $1 - \alpha$ only asymptotically. However, it may fail to achieve it on finite test samples. In real-world decision-making, especially in high-stakes applications, finite test sample coverage guarantees are often essential. Conformal Regression (CR) (Vovk et al., 1999; 2005) provides a principled framework through which PI models can be adapted to ensure such finite-sample coverage guarantees, making them more suitable for practical deployment. A detailed and systematic theoretical treatment of how the conformal framework converts a PI into a valid interval with finite-sample coverage guarantees has been well summarized in (Shafer & Vovk, 2008; Angelopoulos et al., 2024).

Over the last two decades, the literature has witnessed the development of a wide range of UQ algorithms for PI estimation, probabilistic forecasting, and conformal prediction, for trustworthy decision support across diverse application domains. However, the effectiveness of these algorithms largely depends on how the underlying quantile functions are estimated, including both the estimation approach and the learning model employed. In this regard, distribution-free approaches for quantile estimation are generally preferred over distribution-based approaches due to their consistent performance across a wide variety of applications.

With respect to the choice of learning models, the majority of widely adopted UQ algorithms are built upon NN architectures. Representative examples include Quantile Regression Neural Networks (QRNN) (Taylor, 2000; Cannon, 2011), Lower-Upper-Bound Estimation (LUBE) NN (Khosravi et al., 2010), Quality-Driven (QD) loss NN (Pearce et al., 2018), Tube-loss NN (Anand et al., 2026), and Conformalized Quantile Regression (CQR) NN (Romano et al., 2019).

Beyond NN-based models, there is a class of literature that uses Random Forest (RF) for UQ tasks. Meinshausen analyzes the suitability of RF for quantile regression in his work (Meinshausen, 2006), while Lu later established its advantages for PI estimation through a unified error-estimation framework (Lu, 2021). In addition, some recent conformal regression literature prefers to use Quantile Random Forest (QRF) for obtaining the conformal prediction set. Some important examples are (Romano et al., 2019), (Vasiloudis et al., 2019), (Seedat et al., 2023) and (Luo & Zhou, 2025). Apart from the NN and RF models, there is limited recent research literature, such as (Duan et al., 2020) and (Yin et al., 2023), that uses Gradient Boosting (GB) for the UQ task.

Regardless of the learning model employed (NN, RF, or GB), the pair of quantile functions $(f_q(x), f_{1+q-\alpha}(x))$, estimated by them for constructing the PI is inherently subject to estimation uncertainty. Therefore, researchers typically consider two main sources of uncertainty when assessing the overall uncertainty: data uncertainty (aleatoric uncertainty) and model uncertainty (epistemic uncertainty). Data uncertainty arises from the inherent noise or variability in the relationship between $X$ and $Y$ while model uncertainty primarily arises from the uncertainty in fixing the learning model.

Brady Neal identifies the two major sources of model uncertainty in their work (Neal, 2019; Neal et al., 2018), which derives the additive decomposition of model uncertainty into *input uncertainty* and *optimization uncertainty*. The *input uncertainty* quantifies the variation in the learned function across different training sets, while *optimization uncertainty* quantifies the variation in the learned function across multiple training trials on a fixed training set. Most recent UQ literature considers only *optimization uncertainty* for capturing the overall model uncertainty. The ensemble-based approach is a natural method for capturing such uncertainty. Lakshminarayanan et al. (2017) have demonstrated that it is the most effective strategy for capturing model uncertainty in NN and deep learning models. The central idea is to train the same model multiple times over a given training set $T$, using the same hyper-parameter configuration but different random initializations, and to quantify the model uncertainty through the variability observed in the resulting learned parameters or predictions. Furthermore, Pearce et al. (2018) aggregate the model uncertainty captured via ensemble NN with the estimated PIs to quantify the overall uncertainty.

While performing the UQ estimation using the NN, deep learning or RF models, accurately capturing model uncertainty also requires the precise quantification of the *optimization uncertainty*, apart from *input uncertainty*. It is because that all of them exhibit significant *optimization uncertainty* particularly for small

and moderate scale dataset. The objective functions of NN and deep learning models are typically non-convex. This causes the learned solution to oscillate among different local minima across training trials, even under the same hyper-parameter setting and a fixed training set $T$, thereby inducing significant *optimization uncertainty*, particularly when the training set is small. Similar to the NN, RF models also show significant variation in their estimate across training trails as they rely upon the bootstrap sampling and random feature selection at each split.

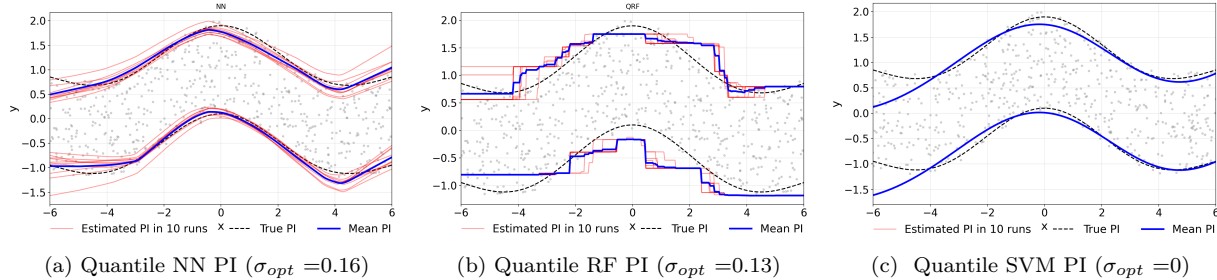

(a) Quantile NN PI ($\sigma_{opt}$ =0.16)  (b) Quantile RF PI ($\sigma_{opt}$ =0.13)  (c) Quantile SVM PI ($\sigma_{opt}$ =0)

Figure 1: The UQ task was to estimate the PI with $1-\alpha = 0.90$ with fixed training set. Unlike SVM, RF and NN based PI model induces the significant variations in estimated PI across 10 independent training runs. This variation can be quantified using the $\sigma_{opt}^2 = \sigma_{0.05}^2 + \sigma_{0.95}^2$, where $\sigma_q^2 = \frac{1}{9k} \sum_{i=1}^{k} \sum_{j=1}^{10} (f_q^{(j)}(x_i) - \bar{f}_q(x_i))^2$, $f_q^{(j)}(x_i)$ and $\bar{f}_q(x_i)$ denoting the $q^{th}$ quantile estimate for $i^{th}$ test point at $j^{th}$ run and their mean over all runs respectively upon $k$ samples. More detailed and related experiments are at Section 5.1.

However, in contrast to NN, deep learning, and RF models, SVMs do not suffer from such *optimization uncertainty*, as they obtain a globally optimal solution that remains invariant across different training trials and independent of initialization under a fixed hyperparameter setting. This makes SVMs less uncertain, more stable, and more trustworthy than their NN and RF counterparts. Beyond global optimality, SVMs also provide simple and often sparse solutions, together with an explicit and principled mechanism for regularization control. Despite these notable advantages, SVMs have remained surprisingly underexplored from the perspective of UQ in regression tasks. However, for classification tasks, a few studies, such as (Wang & Pardalos, 2014; Xiong et al., 2025), have discussed the incorporation of UQ within SVM modeling. In this paper, we re-examine SVMs through the lens of contemporary UQ methodologies and highlight their intrinsic strengths in obtaining stable and less uncertain UQ regression estimates, along with the additional benefits of sparse and simple solutions. We identify the following distinctive and compelling properties of SVM models that strongly support their promotion for UQ regression tasks, especially under small-sample settings.

(a) In the PI estimation task, SVMs fundamentally differ from other existing learning models by exhibiting zero *optimization uncertainty* due to their convergence to a unique globally optimal solution. This makes SVM models exhibit much lower model uncertainty and produce more stable and less uncertain PIs, especially in small-sample learning scenarios (see Figure 1). We also show that NN, RF, and GB models suffer from pronounced model uncertainty as the sample size decreases, which contributes substantially to the overall predictive uncertainty.

(b) SVM models can produce sparse solutions for the lower and upper quantile functions required to construct PIs. The sparsity of the SVM solution not only simplifies the learned quantile functions but, more importantly, also enables effective feature selection in PI estimation tasks. We highlight the significance of feature selection in linear PI estimation tasks and show that SVM models can successfully eliminate a significant percentage of irrelevant features while simultaneously improving PI quality and reducing model complexity, particularly in high-dimensional learning settings.

(c) The adaption of the SVM models in the Conformal Regression (CR) problem can lead to more stable and certain prediction sets without compromising the quality of prediction sets. Surprisingly,

the potential of SVM models for deriving stable CR estimates remains largely unexplored in the literature.

(d) SVM models may be an alternative promising modeling choice for probabilistic forecasting for small and moderate-scale time-series datasets. One direct advantage is that the SVM model yields stable probabilistic forecasts with comparable good-quality PIs, whereas the forecasts of deep learning models vary significantly across training trials, exhibiting substantial *optimization* uncertainty, particularly in the small- and moderate-scale dataset regime.

The remainder of this paper is structured as follows. Section 2 identifies the major sources of model uncertainty and details their quantification. Section 3 describes how SVMs can be used to obtain stable and less uncertain UQ estimates. Section 4 introduces sparse SVM models for PI estimation and demonstrates their use for feature selection in PI estimation tasks. It also extends the SVM framework to obtain stable conformal prediction sets. Section 5 empirically validates our claims regarding the strengths of SVM-based models for UQ tasks. Section 6 concludes the paper and outlines future directions for the adoption of SVM-based UQ methods across diverse application domains.

## 2 Model Uncertainty

Model uncertainty in supervised learning has been studied as the variability of an estimated function under the random choices made during learning. Owing to its significance, this notion has received considerable attention in recent literature, such as (Depeweg et al., 2018; Abdar et al., 2021; Hüllermeier & Waegeman, 2021). In the regression setting, the literature commonly distinguishes two principal sources of uncertainty caused due to this variability and relates them through a single variance identity. We review this decomposition briefly and detail how to quantify them in practice.

Since the estimated function $f$ is subject to significant variation induced by the different stages of the machine learning pipeline, we treat $f$ as a random variable in the subsequent discussion. Let $\mathbf{Var}(f)$ denote the variance of the prediction made by $f$, where $f$ is trained on a finite training set $T$ and evaluated on a held-out test set.

Brady Neel decomposes the total variance $\mathbf{Var}(f)$ into two interpretable sources using the law of total variance in his work (Neal et al., 2018; Neal, 2019). For any random variable $f$ and conditioning variable $Z$, this law states that $\mathbf{Var}(f) = \mathbf{Var}(\mathbb{E}[f \mid Z]) + \mathbb{E}[\mathbf{Var}(f \mid Z)]$, splitting the total variation into the part explained by $Z$ and the part remaining once $Z$ is known. Taking $Z$ to be the training set $T$, we obtain

$$\underbrace{\mathbf{Var}(f)}_{\text{total}} = \underbrace{\mathbf{Var}_T\big(\mathbb{E}[f \mid T]\big)}_{\text{input uncertainty}} + \underbrace{\mathbb{E}_T\big[\mathbf{Var}(f \mid T)\big]}_{\text{optimization uncertainty}} . \tag{1}$$

The first term is the *input uncertainty*, which measures how the average prediction, taken over training trials, varies across different training sets $T$, capturing the model's sensitivity to which data it is trained on. The second term is the *optimization uncertainty*, which measures the variation of the estimated function $f$ across training trials for a fixed $T$, averaged over all training sets. Since the decomposition is an exact instance of the law of total variance, the two terms partition $\mathbf{Var}(f)$ completely, without any assumption of independence between the two sources.

**Practical Quantification of Model Uncertainty:-** Given a training set $T$ and a held-out test set $\{(x_i, y_i) : x_i \in \mathbb{R}^n, y_i \in \mathbb{R}, i = 1, \ldots, k\}$, we empirically quantify the *optimization uncertainty* and *input uncertainty*, which together estimate the overall model uncertainty. Let $f^{(T,j)}$ denote the function learned in the $j^{\text{th}}$ training trial on a fixed training set $T$, then a practical and simpler quantification of the *optimization*

*uncertainty* [1] is obtained through training the model in $M$ independent trials on the fixed given training set $T$ as follows.

$$\sigma_{\text{opt}}^2(f) = \frac{1}{k(M-1)} \sum_{i=1}^{k} \sum_{j=1}^{M} \left( f^{(T,j)}(x_i) - \bar{f}^{(T)}(x_i) \right)^2, \quad \bar{f}^{(T)}(x_i) = \frac{1}{M} \sum_{j=1}^{M} f^{(T,j)}(x_i). \tag{2}$$

To quantify the *input uncertainty* in practice, we draw a sequence of bootstrap subsamples $(T_1, \ldots, T_N)$ of equal size from $T$ and compute

$$\sigma_{\text{in}}^2(f) = \frac{1}{k(N-1)} \sum_{i=1}^{k} \sum_{r=1}^{N} \left( \bar{f}^{(T_r)}(x_i) - \bar{f}(x_i) \right)^2, \text{where } \bar{f}^{(T_r)}(x_i) = \frac{1}{M} \sum_{j=1}^{M} f^{(T_r,j)}(x_i) \text{ and } \bar{f}(x_i) = \frac{1}{N} \sum_{r=1}^{N} \bar{f}^{(T_r)}(x_i) \tag{3}$$

For the PI estimation task with target coverage $1 - \alpha$, a pair of quantile functions $(f_q, f_{1+q-\alpha})$ requires to be estimated for some $0 \leq q \leq \alpha$. In such case, the model uncertainty estimates are given by

$$\sigma_{opt}^2 = \sigma_{opt}^2(f_q) + \sigma_{opt}^2(f_{1+q-\alpha}) \quad \text{and} \quad \sigma_{in}^2 = \sigma_{in}^2(f_q) + \sigma_{in}^2(f_{1+q-\alpha}). \tag{4}$$

Finally, the overall model uncertainty is quantified as

$$\sigma_{model}^2 = \sigma_{opt}^2 + \sigma_{in}^2. \tag{5}$$

## 3 UQ in SVM regression

In this section, we outline SVM methods relevant to UQ estimation techniques in regression tasks.

### 3.1 Quantile Regression and Pinball loss

In distribution-free setting, for a given quantile $q \in (0, 1)$, the quantile value is estimated by minimizing the pinball loss function given by

$$\rho_q(s) = \begin{cases} qs, & \text{if } s \geq 0, \\ (q-1)s, & \text{Otherwise.}, \end{cases} \tag{6}$$

For the estimation of the conditional quantile function, $s$ represents the error obtained by subtracting the estimates $f(x_i)$ from its target values $y_i$. For a given quantile $q \in (0, 1)$, training set $T = \{(x_i, y_i) : x_i \in \mathbf{R}^n, y_i \in \mathbf{R}, i = 1, 2, \ldots m\}$ and class of function $F$, let us suppose that $f_T$ is the solution of the problem $\min_{f \in F} \sum_{i=1}^{m} \rho_q(y - f(x_i))$. Takeuchi et al., have shown that the fraction of $y$ values lying below the function $f_T(x)$ is bounded from above by $qm$ and asymptotically equals $qm$ with probability 1 under a very general condition in their work (Takeuchi et al. (2006)).

Given the training set, SVM models estimate the function in the form of $f(x) = w^T \phi(x) + b$, where $\phi$ maps the input variable $x$ into the high dimensional feature space, such that for any pair of $x_i$ and $x_j$ in $\mathbf{R}^n$, $\phi(x_i)^T \phi(x_j)$ can be obtained by the well defined kernel function $k(x_i, x_j)$. By the use of the kernel trick and representer theorem Schölkopf et al. (2001), the SVM estimate $f(x) = w^T \phi(x) + b$ can be represented by the kernel generated function in the form of $\sum_{i=1}^{m} k(x_i, x) u_i + b$, where $k$ is positive-semi definite kernel Mercer (1909). This representation eliminates the need for explicit knowledge of the mapping $\phi$.

---

[1]This is a practical empirical quantification of the *optimization uncertainty*, in the spirit of the ensemble-based approach of Lakshminarayanan et al. (2017). The exact empirical quantification of (1) additionally requires averaging $\sigma_{\text{opt}}^2$ over multiple training sets, which is subject to the availability of multiple training sets. Even if such sets are generated by bootstrapping from $T$, this would require multiple nested re-trainings of the model when computing the *input uncertainty*, thereby complicating the computation of the *model uncertainty*.

### 3.2   Support Vector Quantile Regression model

The Support Vector Quantile Regression (SVQR) model (Takeuchi et al. (2006)) minimizes the $L_2$-norm of the regularization along with the empirical risk computed by the pinball loss function. For $q^{th}$ quantile function estimation, it seeks the solution of the problem

$$\min_{(w,b)} \frac{1}{2} w^T w + C \sum_{i=1}^{m} \rho_q(y_i - (w^T \phi(x_i) + b)), \tag{7}$$

which can be equivalently converted to the following Quadratic Programming Problem (QPP)

$$\min_{(w,b,\xi,\xi^*)} \frac{1}{2} w^T w + C \sum_{i=1}^{m} (q\xi_i + (1-q)\xi_i^*)$$

$$\text{subject to,}$$
$$y_i - (w^T \phi(x_i) + b) \leq \xi_i,$$
$$(w^T \phi(x_i) + b) - y_i \leq \xi_i^*,$$
$$\xi_i, \xi_i^* \geq 0, \ i = 1, 2, ..m, \tag{8}$$

where $C \geq 0$ is the user defined parameter for trading off the empirical risk against the model complexity.

To efficiently solve QPP (8), we often focus on obtaining the solution to its corresponding Wolfe dual problem, which is given by

$$\min_{(\alpha,\beta)} \sum_{i=1}^{m} \sum_{j=1}^{m} (\alpha_i - \beta_i) k(x_i, x_j)(\alpha_j - \beta_j) - \sum_{i=1}^{m} (\alpha_i - \beta_i) y_i$$

$$\text{subject to,}$$
$$\sum_{i=1}^{m} (\alpha_i - \beta_i) = 0,$$
$$0 \leq \alpha_i \leq Cq, \ i = 1, 2, ..m,$$
$$0 \leq \beta_i \leq C(1-q), \ i = 1, 2, ..m, \tag{9}$$

where $(\alpha_i, \beta_i), i = 1, 2, .., m$, are Lagrangian multipliers.

After obtaining the optimal solution of the dual problem (9), $(\alpha_i^*, \beta_i^*), i = 1, 2, .., m$, the $q^{th}$ quantile function is estimated by

$$f_q(x) = \sum_{i=1}^{m} (\alpha_i^* - \beta_i^*) k(x_i, x) + b. \tag{10}$$

The estimation of the bias term $b$ can be obtained by using the KKT conditions of the primal problem (8). For this, we need to select the every training point $(x_l, y_l)$ which corresponds to $0 < \alpha_l^* < Cq$ or $0 < \beta_l^* < C(1-q)$ and compute

$$b_l = y_l - \sum_{i=1}^{m} (\alpha_i^* - \beta_i^*) k(x_i, x_l). \tag{11}$$

The final value of bias $b$ can be obtained by computing the mean of all $b_l$.

### 3.3   PI estimation in SVM

Given the target confidence $1 - \alpha$ with training set $T$, the PI model requires the estimation of the pair of quantile functions $(f_q(x), f_{1+q-\alpha}(x))$ of the conditional distribution $Y/X$. The SVQR model can be trained twice for the estimation of this pair of quantile functions for some $0 \leq q \leq \alpha$. We detail the algorithm for PI estimation through SVQR in Algorithm 1.

Unlike the NN and RF generated PI, the estimated PI $[f_{\bar{q}}, f_{1+\bar{q}-\alpha}]$ using (1) remains unique and stable for a fixed choice of $C$ and $T$. This is due to the nature of the QPP (9), which always attains a unique global optimal solution.

---

**Algorithm 1** PI estimation through SVQR

---

**procedure** PI THROUGH SVQR($T, 1 - \alpha$)

    Choose $\bar{q} \in [0, \alpha]$

    **for** each $q \in \{\bar{q}, 1 + \bar{q} - \alpha\}$ **do**

        Solve QPP (9) for an appropriate value of $C$.

        Obtain the solution $(\alpha^*, \beta^*)$ and estimate $f_q(x)$.

    **return** $(f_{\bar{q}}, f_{1+\bar{q}-\alpha})$

---

## 4 Sparse PI estimation in SVM

The sparse solution is a yet another promising feature offered by the SVM model. A sparse solution vector in SVM offers significant advantages. It reduces the complexity of the model without compromising its predictive ability and hence expected to own better generalization ability on unseen test points. More importantly, it can help to eliminate the irrelevant features, making the learning task more simple and effective.

### 4.1 Sparse Support Support Vector Quantile Regression

The Sparse Support Support Vector Quantile Regression (SSVQR) can obtain the sparse solution for the quantile estimate. The SSVQR model minimizes the pinball loss function with $L_1$- norm regularization. However, different form of the $L_1$- norm regularization induces different nature of the sparsity leading to different set of benefits.

At first, following the (Mangasarian, 2006), let us express the quantile function $f_q(x) = (w^T \phi(x) + b)$ to be estimated as kernel generated function $\sum_{j=1}^{m} k(x_j, x)u_j + b$, where the expansion coefficients $u \in \mathbb{R}^m$ are the model parameters to be learned. The SSVQR seeks the solution of the following problem

$$\min_{(u,b)} \frac{1}{2}||u||_1 + C \sum_{i=1}^{m} \rho_q \Big( y_i - \big( \sum_{j=1}^{m} k(x_j, x)u_j + b \big) \Big), \tag{12}$$

where $C \geq 0$ is the user defined parameter for trading of the regularization against the empirical loss. Also, the $\rho_q(s)$ is the pinball loss function as defined in (6) that can be equivalently expressed as $q \max(s, 0) + (1 - q) \max(-s, 0)$. Further, let us consider $s_i = y_i - \big( \sum_{j=1}^{m} k(x_j, x_i)u_j + b \big)$ and two slack variables $\xi_i = \max(s_i, 0)$ and $\xi_i^* = \max(-s_i, 0)$ to measure the error corresponding to the data points lying above and lower side of the regression function respectively. Now, the problem (12) can be written as

$$\min_{(u,b)} \frac{1}{2}||u||_1 + C \sum_{i=1}^{m} (q\xi_i + (1 - q)\xi_i^*),$$

$$\text{subject to,}$$

$$\xi_i = \max \Big( y_i - \big( \sum_{j=1}^{m} k(x_j, x_i)u_j + b \big), 0 \Big), i = 1, 2, ...m,$$

$$\xi_i^* = \max \Big( \big( \sum_{j=1}^{m} k(x_j, x_i)u_j + b \big) - y_i, 0 \Big), i = 1, 2, ...m,$$

that can be simplified as

$$\min_{(u,b,\xi,\xi^*)} \frac{1}{2}||u||_1 + C\sum_{i=1}^{m}(q\xi_i + (1-q)\xi_i^*)$$

subject to,

$$\xi_i \geq y_i - \Big(\sum_{j=1}^{m} k(x_j,x_i)u_j + b\Big), \ \ \xi_i \geq 0, \ i = \ 1,2,..m,$$

$$\xi_i^* \geq \Big(\sum_{j=1}^{m} k(x_j,x_i)u_j + b\Big) - y_i, \ \ \xi_i^* \geq 0, \ i = \ 1,2,..m,$$

which can be written as

$$\min_{(u,b,\xi,\xi^*)} \frac{1}{2}||u||_1 + C\sum_{i=1}^{m}(q\xi_i + (1-q)\xi_i^*)$$

subject to,

$$y_i - \Big(\sum_{j=1}^{m} k(x_j,x_i)u_j + b\Big) \leq \xi_i,$$

$$\Big(\sum_{j=1}^{m} k(x_j,x_i)u_j + b\Big) - y_i \leq \xi_i^*,$$

$$\xi_i,\xi_i^* \geq 0, \ i = \ 1,2,..m. \tag{13}$$

Without loss of generality, let us consider the solution vector $u = r - p$, where $r$ and $p$ are vectors of positive numbers i,e., $r_i, p_i > 0, i = 1, 2, .., m$ , then similar to the Mangasarian (2006), the problem (13) can be expressed as

$$\min_{(r,p,b,\xi,\xi^*)} \frac{1}{2}\sum_{i=1}^{m}(r_i + p_i) + C\sum_{i=1}^{m}(q\xi_i + (1-q)\xi_i^*)$$

subject to,

$$y_i - \Big(\sum_{j=1}^{m} k(x_j,x_i)(r_j - p_j) + b\Big) \leq \xi_i,$$

$$\Big(\sum_{j=1}^{m} k(x_j,x_i)(r_j - p_j) + b\Big) - y_i \leq \xi_i^*,$$

$$\xi_i,\xi_i^*,r_i,p_i \geq 0, \ i = \ 1,2,..m. \tag{14}$$

The above problem (14) is a LPP with $4m$ variables, $2m$ linear constraints and $4m$ non-negative constraints, which can be efficiently solved by any LPP solver. The optimal solution $(r^*, p^*, b^*)$ of the LPP (14) determines the solution $(u^*, b^*)$ of the problem (12), leading to the estimation of the $q^{th}$ quantile function as

$$f_q(x) = \sum_{i=1}^{m} k(x_i,x)u_i^* + b^*. \tag{15}$$

One important advantages of the SSVQR model detailed above is that the solution vector $(u^*, b^*)$ remains sparse, similar to the solution of problem (24) of (Mangasarian (2006)). It simplifies the estimated quantile function such that it is constructed through the kernel of few training points, often leading to better quality estimate. However, such sparsity can not help to eliminate the irrelevant features of $x$, even while learning a linear quantile function using the linear kernel function.

To carry out the feature selection task while obtaining the $q^{th}$ linear quantile estimate, we may consider the optimization problem similar to the LASSO problem in quantile estimation (Li & Zhu, 2008) as follows.

$$\min_{(w_q,b_q)} \frac{1}{2}||w_q||_1 + C\sum_{i=1}^{m} \rho_q\Big(y_i - (w_q^T x + b_q)\Big), \tag{16}$$

that can be converted into the LPP similar to the problem (12) as follows.

$$\min_{(r^{(1)}, p^{(1)}, b, \xi, \xi^*)} \frac{1}{2} \sum_{i=1}^{n} (r_i^{(1)} + p_i^{(1)}) + C \sum_{i=1}^{m} (q\xi_i + (1-q)\xi_i^*)$$

subject to,

$$y_i - \left( (r^{(1)} - p^{(1)})^T x_i + b \right) \le \xi_i,$$

$$\left( (r^{(1)} - p^{(1)})^T x_i + b \right) - y_i \le \xi_i^*,$$

$$r^{(1)}, p^{(1)} \ge 0, \quad \xi_i, \xi_i^* \ge 0, \ i = 1, 2, ..m, \tag{17}$$

where $r^{(1)}$ and $p^{(1)}$ are $n$-dimensional positive vectors such that $w_q$ can be expressed as $r^{(1)} - p^{(1)}$.

The asymptotical properties of the SSVQR model remains immune in view of the Lemma 3 of (Takeuchi et al. (2006)) which holds for a wide range of regularization schemes used in pinball-loss minimization within the SVM framework.

### 4.2 PI estimation through SSVQR

The PI estimation through SSVQR is straightforward on the line of SVQR. The two quantile bound functions $f_{\bar{q}}(x)$ and $f_{1+\bar{q}-\alpha}(x)$ should be estimated by solving a pair of LPP problems (14) with $q = \bar{q}$ and $q = 1+\bar{q}-\alpha$ respectively. The solution of LPP problem (14) is sparse that makes the estimated quantile functions $f_{\bar{q}}(x)$ and $f_{1+\bar{q}-\alpha}(x)$ more simple that may enhance their generalization ability.

### 4.3 Feature selection in PI estimation through Sparse SVM

Now, we address the significance of the feature selection in UQ regression task that has also been unexplored in the UQ literature. Similar to other machine learning tasks, the PI estimation in high-dimensional settings also presents several challenges. The increased dimensionality not only increases the complexity of the PI bounds but also necessitates a larger sample size to ensure the quality of the estimate PI. Therefore, an efficient feature selection method is crucial for reducing the overall complexity of the PI estimation, particularly when dealing with high-dimensional data.

---

**Algorithm 2** Feature selection through SSVQR

---

**procedure** :- FEATURE SELECTION THROUGH SSVQR($T, 1 - \alpha, tol$)

    Choose some $\bar{q} \in [0, \alpha]$

    **for** each $q \in \{\bar{q}, (1 + \bar{q} - \alpha)\}$ **do**

        Solve the LPP (17) and obtain its solution $(r^{(1)*}, p^{(1)*}, b^*)$.

        Obtain the $w_q$ as $(r^{(1)*} - p^{(1)*})$.

    Compute $I_{\bar{q}} = \{i :, |w_{\bar{q}}(i)| \le tol\}$ and $I_{1+\bar{q}-\alpha} = \{i :, |w_{1+\bar{q}-\alpha}(i)| \le tol\}$

    Compute $I = I_{\bar{q}} \cap I_{1+\bar{q}-\alpha}$ and *Feature Set* $= \{1, 2, ..., m\} - I$

    **return** (*Feature Set*)

---

We detail the feature selection algorithm through SSVQR model for the linear PI estimation task in Algorithm 3. Here, a linear PI refers to the PI, where both bounds are linear functions of the input variables ,i,e. $f_{\bar{q}}(x) = w_{\bar{q}}^T x + b_{\bar{q}} \ f_{1+\bar{q}-\alpha}(x) = w_{1+\bar{q}-\alpha}^T x + b_{1+\bar{q}-\alpha}$.

### 4.4 Conformal Regression in SVM

Further, we extend the SVM based PI model in CR setting for obtaining the finite sample test set guarantees. In split CR setting, we detail the SVM based CR algorithm using the non-conformity score of (Romano et al., 2019) as follows.

---

**Algorithm 3** CR through SVQR

---

**procedure** CR THROUGH SVQR($T$, $1 - \alpha$)

    Split $T$ into training $I_1$ and calibration $I_2$

    Choose $\bar{q} \in [0, \alpha]$

    **for** $q \in \{\bar{q},\ 1 + \bar{q} - \alpha\}$ **do**

        Train SVQR by solving QPP (9) on $I_1$ to obtain $f_q$

    $E_i \leftarrow \max\{f_{\bar{q}}(x_i) - y_i,\ y_i - f_{1+\bar{q}-\alpha}(x_i)\},\ x_i \in I_2$

    $Q \leftarrow (1 - \alpha)(1 + |I_2|^{-1})$-quantile of $\{E_i\}$

    **return** $[\,f_{\bar{q}}(x) - Q,\ f_{1+\bar{q}-\alpha}(x) + Q\,]$

---

The prediction set produced by Algorithm 3 remains stable across training trials without compromising its quality, whereas RF and NN models produce CR prediction sets that exhibit significant variability across different training runs even under the same hyper-parameter setting.

## 5 Experimental Results

In this section, we present the numerical results to verify our claims regarding the usefulness of the SVM in UQ regression tasks.

### 5.1 SVMs are less uncertain for PI estimation

To study the characteristics of SVM relative to counterpart learning models such as NN, RF, and GB, we simulate the simple artificial dataset of Figure 1. We generate $2m$ training points and $k$ test points according to $y_i = \frac{\sin(x_i)}{x_i} + \epsilon_i$, where $\epsilon_i \sim \text{Uniform}(-1, 1)$. The task is to construct a PI with a target coverage of 0.90. Since the noise distribution is symmetric, the narrowest valid PI is obtained by estimating the $0.05^{th}$ and $0.95^{th}$ conditional quantile functions, $f_{0.05}(x)$ and $f_{0.95}(x)$. The analytically optimal quantile functions, derived by inverting the uniform cumulative distribution function, serve as the ground-truth PI and are shown as black dotted lines in Figure 1.

To quantify the data uncertainty, we measure how much the quantile estimates produced by a learning model deviate from the true quantile functions on the test set. Specifically, we compute the Mean of RMSEs of the two estimated quantile functions,

$$\text{Mean of RMSEs} := \frac{1}{2}\left(\sqrt{\frac{1}{k}\sum_{i=1}^{k}\left(f_{0.05}(x_i) - \hat{f}_{0.05}(x_i)\right)^2} + \sqrt{\frac{1}{k}\sum_{i=1}^{k}\left(f_{0.95}(x_i) - \hat{f}_{0.95}(x_i)\right)^2}\right), \quad (18)$$

where $\hat{f}_q(x)$ denotes the estimate of the $q^{th}$ quantile function obtained by a learning model.

In addition, we compute the PICP, MPIW, and Interval Score (IS) (Winkler, 1972) to assess the quality of the PI estimates. However, these metrics capture only the global quality of a PI and may not reflect its local correctness. In contrast, the mean of RMSEs quantifies the correctness of the PI both globally and locally.

To study model uncertainty, we quantify the *optimization uncertainty* and *input uncertainty* using (20) and (3), respectively. For $\sigma_{\text{opt}}^2$, we set $M = 10$ in (20) and obtain it from (4). For $\sigma_{\text{in}}^2$, we set $N = 10$ in (3) and generate 10 different full training sets of size $m$ by varying the random seed and obtain it from (4)

The next step is to tune the parameters of each learning model. For this, we hold out 50% of the training points as a validation set; the test points are never revealed to the model during either the training or the tuning phase. Actually only $m$ data points were used for training and $m$ data points were kept separately for the validation purpose. We carefully and impartially tuned the parameters of each learning model. For NN, we use a single hidden layer of 512 neurons. All such hyper-parameter choices for each learning model are justified in Appendix A, along with a detailed ablation study.

| | | Data uncertainty | | | | Model uncertainty | | |
|---|---|---|---|---|---|---|---|---|
| Model | $m$ | MRMSE | PICP | MPIW | IS | $\sigma_{\mathrm{opt}}$ | $\sigma_{\mathrm{in}}$ | $\sigma_{\mathrm{model}}$ |
| ANN | 50 | 0.2034 | 0.8235 | 1.6274 | 2.2457 | 0.1578 | 0.2720 | 0.3145 |
| | 100 | 0.1742 | 0.9178 | 1.9174 | 2.1133 | 0.3527 | 0.2131 | 0.4121 |
| | 200 | 0.0981 | 0.9223 | 1.8470 | 1.9721 | 0.2462 | 0.1375 | 0.2820 |
| | 300 | 0.0849 | 0.9242 | 1.8453 | 1.9552 | 0.2060 | 0.1246 | 0.2408 |
| | 400 | 0.0797 | 0.9260 | 1.8411 | 1.9489 | 0.1817 | 0.1067 | 0.2107 |
| | 800 | 0.0541 | 0.9175 | 1.8118 | 1.9190 | 0.1485 | 0.0789 | 0.1682 |
| QRF | 50 | 0.2206 | 0.8587 | 1.7645 | 2.2427 | 0.1237 | 0.3171 | 0.3404 |
| | 100 | 0.1592 | 0.8537 | 1.7063 | 2.1558 | 0.0880 | 0.2262 | 0.2427 |
| | 200 | 0.1593 | 0.8430 | 1.6715 | 2.1139 | 0.0736 | 0.2136 | 0.2260 |
| | 300 | 0.1518 | 0.8442 | 1.6633 | 2.1045 | 0.0669 | 0.2005 | 0.2114 |
| | 400 | 0.1486 | 0.8543 | 1.6800 | 2.0952 | 0.0597 | 0.1909 | 0.2000 |
| | 800 | 0.1499 | 0.8478 | 1.6625 | 2.0828 | 0.0613 | 0.1913 | 0.2009 |
| **SVM** | 50 | 0.1995 | 0.8000 | 1.5653 | 2.2582 | **0.0** | 0.2529 | **0.2529** |
| | 100 | **0.1064** | 0.8753 | 1.7155 | 2.0138 | **0.0** | 0.1405 | **0.1405** |
| | 200 | **0.0726** | 0.8928 | 1.7555 | 1.9413 | **0.0** | 0.1028 | **0.1028** |
| | 300 | **0.0593** | 0.8987 | 1.7688 | 1.9277 | **0.0** | 0.0798 | **0.0798** |
| | 400 | **0.0549** | 0.9033 | 1.7792 | 1.9234 | **0.0** | 0.0754 | **0.0754** |
| | 800 | **0.0391** | 0.9088 | 1.7858 | 1.9039 | **0.0** | 0.0570 | **0.0570** |
| XGBoost | 50 | 0.3480 | 0.7022 | 1.4064 | 2.9606 | 0.0641 | 0.3185 | 0.3249 |
| | 100 | 0.2245 | 0.7858 | 1.5536 | 2.3224 | 0.0506 | 0.1924 | 0.1989 |
| | 200 | 0.1599 | 0.8195 | 1.6135 | 2.1051 | 0.0315 | 0.1546 | 0.1578 |
| | 300 | 0.1319 | 0.8368 | 1.6460 | 2.0293 | 0.0292 | 0.1387 | 0.1418 |
| | 400 | 0.1160 | 0.8545 | 1.6785 | 1.9916 | 0.0230 | 0.1243 | 0.1264 |
| | 800 | 0.0791 | 0.8728 | 1.7182 | 1.9351 | 0.0183 | 0.0919 | 0.0937 |
| NGB | 50 | 0.2688 | 0.8470 | 1.7539 | 2.3869 | **0.0** | 0.3623 | 0.3623 |
| | 100 | 0.2063 | 0.8698 | 1.8090 | 2.2214 | **0.0** | 0.2822 | 0.2822 |
| | 200 | 0.1818 | 0.9155 | 1.9048 | 2.1200 | **0.0** | 0.2287 | 0.2287 |
| | 300 | 0.1765 | 0.9327 | 1.9419 | 2.0898 | **0.0** | 0.1979 | 0.1979 |
| | 400 | 0.1714 | 0.9457 | 1.9808 | 2.0841 | **0.0** | 0.1657 | 0.1657 |
| | 800 | 0.1669 | 0.9572 | 2.0086 | 2.0718 | **0.0** | 0.1277 | 0.1277 |
| GPR | 50 | **0.1692** | 0.8692 | 1.7354 | 2.1265 | **0.0** | 0.2578 | 0.2578 |
| | 100 | 0.1297 | 0.8582 | 1.6962 | 2.0334 | **0.0** | 0.1778 | 0.1778 |
| | 200 | 0.0961 | 0.8730 | 1.7145 | 1.9661 | **0.0** | 0.1366 | 0.1366 |
| | 300 | 0.0972 | 0.8663 | 1.7018 | 1.9624 | **0.0** | 0.1297 | 0.1297 |
| | 400 | 0.0829 | 0.8685 | 1.7023 | 1.9530 | **0.0** | 0.1078 | 0.1078 |
| | 800 | 0.0737 | 0.8668 | 1.7032 | 1.9325 | **0.0** | 0.0866 | 0.0866 |

Table 1: Uncertainty estimates obtained by different learning models across training sizes $m$. The left block reports data-uncertainty metrics; the right block reports the model-uncertainty decomposition $\sigma_{\mathrm{model}} = \sqrt{\sigma_{\mathrm{opt}}^2 + \sigma_{\mathrm{in}}^2}$. SVM, GPR, and NGB are deterministic ($\sigma_{\mathrm{opt}} = 0$), IS:- Interval Score (Winkler, 1972)

We compare the performance of SVM against several available learning models, including NN, QRF, Gaussian process regression (GPR), and two boosting variants, namely eXtreme Gradient Boosting (XGBoost) (Chen & Guestrin, 2016) and Natural Gradient Boosting (NGB) (Duan et al., 2020), in terms of data uncertainty and model uncertainty in Table 1. All models are evaluated on a test set of size $k = 600$ and trained on training sets of varying size. We visualize these results in Figures 2 and 3.

Figure 2(a) shows that the quality of the PI improves with training set size for every learning model in a similar fashion. However, SVM tends to produce higher-quality PIs, and hence lower data uncertainty, than the other models. Figure 3 plots the model uncertainty of each model against training set size. The *optimization uncertainty* (Fig. 3(a)) of GPR, NGB and SVM remains zero, while XGBoost exhibits lower optimization uncertainty than RF and NN; in all cases it decreases as the number of training samples grows. The *input uncertainty* (Fig. 3(b)) decreases almost exponentially with training set size for every model, with SVM showing the least input uncertainty overall. XGBoost and RF remain most unstable under changes of

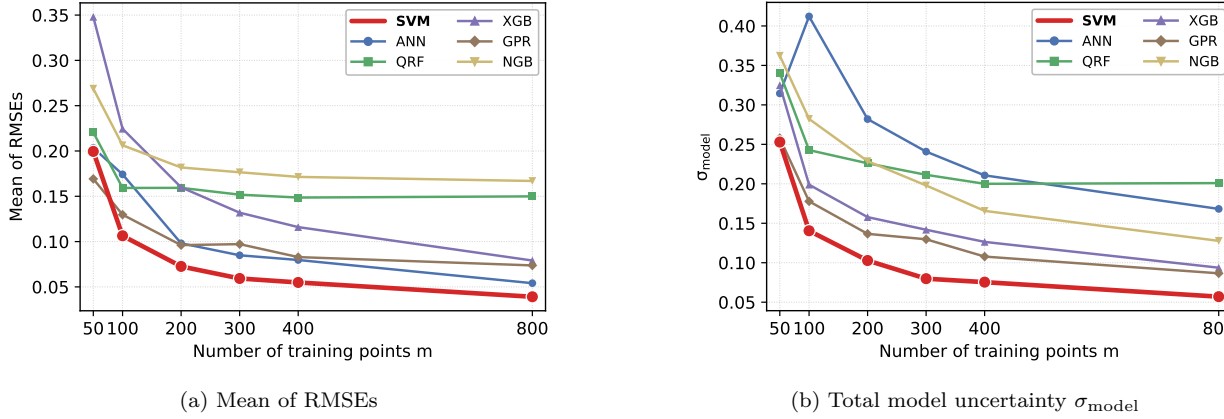

(a) Mean of RMSEs

(b) Total model uncertainty $\sigma_{\mathrm{model}}$

Figure 2: Predictive quality and total model uncertainty as a function of the training size $m$. SVM (bold red) attains the lowest values across all training sizes.

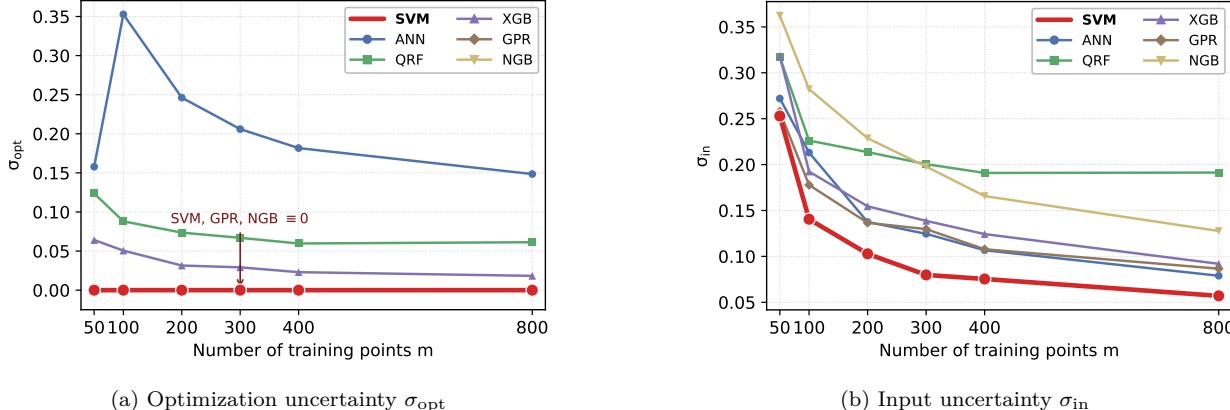

(a) Optimization uncertainty $\sigma_{\mathrm{opt}}$

(b) Input uncertainty $\sigma_{\mathrm{in}}$

Figure 3: Decomposition of model uncertainty into its optimization and input components as a function of the training size $m$. SVM, GPR, and NGB are deterministic, so $\sigma_{\mathrm{opt}} = 0$ for all three.

the training set and hence exhibit high *input uncertainty*. SVM obtains least overall model uncertainty as shown in Figure 2(b) among all used learning models. SVM seems to be the most promising learning model for estimation of the both data uncertainty and model uncertainty.

Building on the numerical results of Table 1, illustrated in Figures 2 and 3, we compare SVM against each of its counterpart learning models below.

(i) **SVM vs. NN:-** SVM tends to produce better overall uncertainty estimates than NN. From the model-uncertainty perspective, comprising both *optimization* and *input uncertainty*, SVM is a superior and more stable choice than NN as NN exhibits significant *optimization uncertainty*, particularly at small training sizes, and its *input uncertainty* is also larger than that of SVM. This advantage is expected to widen in high-dimensional settings where only a small to moderate number of samples is available.

(ii) **SVM vs. QRF:-** QRF yields poorer PI estimates than the other models. Because its training relies on bootstrap sampling, it exhibits substantial *optimization uncertainty*, and it is also the most unstable model under variation of the training set, with correspondingly high *input uncertainty*.

(iii) **SVM vs. XGBoost:-** XGBoost exhibits very low *optimization uncertainty* and remains nearly stable across training trials; in fact, it becomes fully deterministic, with zero *optimization uncertainty*, when its parameter subsampling ratio is set to one. In our experiments, we tune this ratio

| Dataset | Dim. | Before FS | | | After FS | | | %Red. |
|---|---|---|---|---|---|---|---|---|
| | | PICP | MPIW | Time | PICP | MPIW | Time | Feature. |
| Spambase | (4601,58) | 0.966 | 0.933 | 60 | 0.965 | 0.934 | 52 | 46% |
| Student Perf. | (395,16) | 0.886 | 6.843 | 0.48 | 0.886 | 6.843 | 0.42 | 73% |
| Boston Housing | (505,14) | 0.941 | 23.06 | 0.81 | 0.941 | 23.06 | 0.70 | 38% |
| UCI-SECOM | (1568,591) | 0.876 | 1.615 | 16 | 0.920 | 1.668 | 7 | 91% |
| MADELON | (2000,500) | 0.710 | 2.001 | 58 | 0.938 | 2.002 | 11 | 99% |

Table 2: Performance before and after feature selection using Algorithm 2. Time is overall runtime in seconds. FS: Feature Selection

to 0.8, sampling 80% of the data points for each tree, which improves PI quality. From the model uncertainty perspective, however, a key drawback of XGBoost is that its *input uncertainty* is more pronounced than that of SVM and NN, since the greedy, tree-based partitioning is highly sensitive to the choice of training set. Also, from the data uncertainty perspective, both SVM and NN produce higher-quality PIs than XGBoost as reflected in Figure 2.

(iv) **SVM vs. GPR:-** Like SVM, GPR is fully deterministic across training trials and therefore exhibits zero *optimization uncertainty*; its overall learning mechanism is closely related to that of the least-squares SVM (Suykens et al., 2002) However, GPR assumes a normal noise distribution when constructing the PI, which limits its quality under non-normal noise, as clearly reflected in our numerical results, also illustrated through Figure 2.

(v) **SVM vs. NGB.** Like SVM, NGB Duan et al. (2020) is fully deterministic and therefore always attains zero *optimization uncertainty*. However, it exhibits the most pronounced *input uncertainty* among all models, which makes its overall model uncertainty substantial. From the data-uncertainty perspective, NGB also produces poor-quality PI estimates. In contrast to the quantile-based SVM, NGB is a distribution-based model that relies on a specific parametric assumption on the noise, which limits its ability to adapt to a variety of noise distributions.

Furthermore, we find that the estimates of data uncertainty (PI quality) and model uncertainty (comprising *input uncertainty* and *optimization uncertainty*) do not depend heavily on the test set size. In contrast, they depend strongly on the training set size, as evident in Figure 2. Appendix A.4 provides detailed experiments showing that the data uncertainty and model uncertainty estimates do not depend strongly on the test set size.

## 5.2 Feature selection is important in PI estimation task

Next, we apply our Sparse SVM model for feature selection in PI estimation with target coverage $1-\alpha = 0.95$ and linear kernel following the Algorithm 2.

Experimental Setup:- All of the simulations were performed in the MATLAB. We use the 80% of data points for training and use rest of them for testing. Test points were remained untouched during training.

Datasets:- we use five popular real-world benchmark datasets namely Spambase $(4601 \times 58)$ (Hopkins & Suermondt (1999)), Student Performance $(395 \times 16)$ (Cortez (2008)) , Boston Housing $(505 \times 14)$ (Harrison Jr & Rubinfeld (1978)), UCI-secom $(1568 \times 591)$ (McCann & Johnston (2008)) and MADELON $(2000 \times 500)$ (Guyon (2004)) for the feature selection task using our proposed algorithm.

Following Algorithm 2 , we perform feature selection using the SSVQR PI model for $\bar{q} = 0.025$ and present the results in Table 2. Since Algorithm 2 solves the problem (14) with linear kernel twice, we need to tune only the parameter $C$ for each $\bar{q}$ which was selected from the grid of $\{2^i : i = -15, ...., 15\}$. We select the value of $C$ that provides the required quantile coverage upon the training set.

On average, Algorithm 2 could reduce the 69% of features on considered datasets and reduce the complexity of the PI estimation task significantly, while maintaining the quality of the PI, as measured by PICP and MPIW. The training time of the PI estimating task improves around 53% after dropping irrelevant features through Algorithm 2. Also, it leads to the reduction of tuning and testing time of the PI model. For high-dimensional datasets such as UCI secom and MADELON, feature selection leads to a significant improvement in the quality of the estimated PI in terms of its coverage. Table 3 lists the dropped features by Algorithm 2 in case of different datasets.

| Dataset | Dropped Features |
|---|---|
| Spambase | 2, 8, 11, 13, 16, 17, 18, 19, 20, 21, 22, 23, 25, 27, 31, 35, 37, 46, 47, 48, 49, 50, 53, 54, 55, 56 |
| Student Perf. | 1, 2, 3, 4, 5, 6, 7, 8, 9, 10, 11 |
| Boston Housing | 2, 3, 4, 7, 10 |
| UCI Secom | 1, 4-20, 22-27, 28-41, 42-50, 52-58, 60-66, 68-70, 72, 74-87, 89, 91-110, 112-132, 134, 137-138, 140-157, 159-160, 162-187, 189-203, 205-224, 226-246, 247-296, 297-332, 333-362, 364-386, 387-400, 401-418, 420-422, 424-431, 434-435, 437-438, 440-466, 467, 469-481, 483, 489-498, 501-509, 512-520, 522-538, 540-545, 546, 548-560, 563-569, 571, 573-579, 580, 582-587 |
| MADELON | 0-89, 91-227, 229-275, 277-331, 333-444, 446-499 |

Table 3: Features dropped by the Algorithm 2. On average, it drops the 69% of features on considered datasets and reduce the complexity of the PI estimation task significantly.

## 5.3 SVM can obtain more stable conformal prediction sets

We compare the SVM-based CR model detailed in Algorithm 3 against the NN-based CR model (NN+CQR) Romano et al. (2019), the RF-based CR model (QRF+CR) Romano et al. (2019), and the XGBoost-based CR model (XGBoost+CR) on popular benchmark datasets. The SVM-based CR model often achieves low model uncertainty while still producing a good-quality prediction set. Moreover, the SVM-based CR model is stable, as it obtains the same CR prediction set across different training trials.

We report the numerical results characterizing the data uncertainty and model uncertainty obtained by each learning model in Table 4. Each learning model was required to obtain a PI with target coverage $1-\alpha = 0.90$. Each dataset was partitioned into a training set (60%), a calibration set (20%), and a test set (20%). All models were first trained to estimate the $0.05^{\text{th}}$ and $0.95^{\text{th}}$ quantile functions, after which the nonconformity scores used in Romano et al. (2019) were computed for conformal prediction.

We first carefully tune the parameters of each model to obtain the best-quality PI without touching the test set. The best hyper-parameters obtained for each learning model are listed in Table 10. All hyper-parameters were tuned using grid search with the appropriate search space, also detailed at the Appendix A.2

To measure the *optimization uncertainty*, which characterizes the stability of the model, we train learning models across $M = 10$ independent runs using fixed hyper-parameter settings and identical splits of the training and calibration sets, and compute the variance of each quantile function on a fixed test set using (20) and aggregate them through (4). To quantify the *input uncertainty*, we use (3) with $M = 10$ and $N = 10$ to measure the overall variance of the estimated quantile functions across 10 different random splits of the dataset into training, calibration, and test sets, using the same ratio and hyper-parameter settings.

The results in Table 4 show that out of five datasets, SVM based CR model obtains best quality prediction set along with least model uncertainty on three datasets. On the Concrete dataset, it also manages a good balance between the model uncertainty and data uncertainty estimate compared to the other learning models. While the NN-based model produces high-quality CR prediction set on Concrete dataset, it exhibits relatively high model uncertainty. In contrast, the XGBoost model yields lower model uncertainty but comparatively poorer CR prediction set. The SVM model lies between these two extremes, offering a balanced trade-off between PI quality and model uncertainty, which makes it promising choice for CR modeling on Concrete dataset also.

The estimates of the SVM and XGBoost models remain stable across trials, with zero *optimization uncertainty*. The SVM model's estimate is inherently fully deterministic, whereas the XGBoost estimate is deterministic only with the parameter *subsample* = 1, which we use in our experiments. In such case, each boosting iteration uses the entire training dataset rather than a random subset of samples.

| Dataset | Method | PICP | MPIW | $\sigma_{\mathbf{opt}}$ | $\sigma_{\mathbf{in}}$ | $\sigma_{\mathbf{model}}$ | Time (s) |
|---|---|---|---|---|---|---|---|
| Boston $(506 \times 13)$ | CQR-NN | 0.94 | 10.01 | 2.61 | 3.00 | 3.98 | 6.34 |
| | SVQR+CR | 0.91 | **9.60** | 0.00 | 1.96 | **1.96** | 0.23 |
| | QRF+CR | 0.91 | 11.01 | 1.45 | 2.25 | 2.68 | 0.11 |
| | XGBoost+CR | 0.92 | 11.84 | 0.00 | 2.31 | 2.31 | 1.02 |
| Concrete $(1030 \times 8)$ | CQR-NN | 0.92 | **17.84** | 4.41 | 2.55 | 5.09 | 3.29 |
| | SVQR+CR | 0.92 | 20.39 | 0.00 | 3.62 | 3.62 | 6.42 |
| | QRF+CR | 0.91 | 20.42 | 2.39 | 3.29 | 4.07 | 0.23 |
| | XGBoost+CR | 0.90 | 22.07 | 0.00 | 3.45 | **3.45** | 0.20 |
| AutoMPG $(398 \times 7)$ | CQR-NN | 0.94 | 9.77 | 2.23 | 1.47 | 2.67 | 1.14 |
| | SVQR+CR | 0.91 | **9.18** | 0.00 | 1.66 | **1.66** | 0.19 |
| | QRF+CR | 0.94 | 9.69 | 1.43 | 1.78 | 2.28 | 0.07 |
| | XGBoost+CR | 0.93 | 9.84 | 0.00 | 2.07 | 2.07 | 0.40 |
| RealEstate $(414 \times 6)$ | CQR-NN | 0.92 | 23.32 | 2.46 | 3.60 | 4.36 | 0.46 |
| | SVQR+CR | 0.92 | **21.71** | 0.00 | 3.06 | **3.06** | 0.28 |
| | QRF+CR | 0.95 | 23.58 | 2.90 | 5.90 | 6.57 | 0.19 |
| | XGBoost+CR | 0.93 | 23.56 | 0.00 | 4.62 | 4.62 | 0.10 |
| Servo $(167 \times 4)$ | CQR-NN | 0.98 | **1.52** | 0.25 | 0.29 | **0.38** | 0.10 |
| | SVQR+CR | 0.98 | 1.87 | 0.00 | 0.38 | **0.38** | 0.06 |
| | QRF+CR | 0.95 | 1.61 | 0.28 | 0.58 | 0.64 | 0.14 |
| | XGBoost+CR | 0.85 | 2.86 | 0.00 | 0.78 | 0.78 | 0.09 |

Table 4: Conformal prediction sets obtained by different learning models across benchmark regression datasets. $\sigma_{\mathrm{model}} = \sqrt{\sigma_{\mathrm{opt}}^2 + \sigma_{\mathrm{in}}^2}$ denotes the overall model uncertainty. On three datasets, the SVM manages to obtain the best PI quality and model uncertainty estimate simultaneously. On the Concrete dataset, the SVM model achieves a favorable balance between data uncertainty, reflected through PI quality, and model uncertainty, compared with the other learning models. The estimates of the SVM and XGBoost models remain stable across training trials, with zero *optimization uncertainty*. The SVM model's estimate is inherently fully deterministic, whereas the XGBoost estimate is deterministic only with the parameter *subsample* = 1, which we use in our experiments. The tuned hyper-parameters for each learning model are listed in Table 10 of the Appendix A.

## 5.4 SVM potential for probabilistic forecasting

In this section, we perform the experiments to reveal the potential of SVM for probabilistic forecasting task particularly for small and moderate scale datasets. One advantage of the SVM model over modern deep learning model its stability of the prediction leading to much lower model uncertainty. However, the practical advantages of the SVM seems to be diminishing as the size of the training set grows.

### 5.4.1 Benchmark Datasets

We compare the quality of the probabilistic forecasting obtained by SVM models against the recently developed complex deep probabilistic forecasting architectures upon popular time-series datasets namely Female Births $(365 \times 1)$ (Datamarket.com), Minimum Temperature $(3651 \times 1)$ (machinelearningmastery.com) and Beer Production $(464 \times 1)$ (Australian (1996)). We have used the 70% of dataset as training set and rest of them are used for testing. Out of the training set, the last 10% of the observations have been used for the validation set.

Baseline Methods:- We also train several recent and widely adopted Quantile based deep learning architectures for probabilistic forecasting developed in distribution-free setting, including Quantile-based LSTM, Quantile-based GRU, Quantile-based TCN, Quantile-based Transformers and Quantile based Mamba (Gu & Dao, 2023) models. The SVQR and SSVQR models are fundamentally quantile-based probabilistic forecasting approaches; therefore, when comparing them with deep learning counterparts, we should only consider quantile-based deep learning methods. However, for the sake of extensive comparisons, we also report the per-

| Data (Size) | Method | PICP | MPIW | $\sigma^2_{opt}$ | #W | Sparsity (%) |
|---|---|---|---|---|---|---|
| FB | SSVQR | 0.93 | 28.00 | 0 | 256 | 61 |
| | SVQR | **0.95** | **27.11** | **0** | 256 | 0 |
| | Quantile-LSTM | 0.95 | 28.20 | 1.52 | 30K | 0 |
| | Quantile-GRU | 0.95 | 46.52 | 0.39 | 65K | 0 |
| | Quantile-TCN | 0.96 | 57.00 | 5.54 | 120K | 0 |
| | Quantile-Transformer | 0.95 | 59.00 | 17.34 | 80K | 0 |
| | Quantile-Mamba | 0.94 | 26.91 | 0.42 | 30K | 0 |
| | Tube-LSTM | 0.96 | 28.09 | 0.34 | 30K | 0 |
| | QD-LSTM | 0.94 | 38.98 | 2.50 | 30K | 0 |
| | DeepAR | 0.94 | 29.80 | 0.14 | 13K | 0 |
| MnTemp | SSVQR | 0.96 | 10.72 | **0** | 2556 | 69 |
| | SVQR | 0.96 | 75.81 | 0 | 2556 | 0 |
| | Quantile-LSTM | 0.95 | 24.82 | 0.14 | 32K | 0 |
| | Quantile-GRU | 0.94 | 16.53 | 0.19 | 65K | 0 |
| | Quantile-TCN | 0.95 | 16.84 | 0.18 | 120K | 0 |
| | Quantile-Transformer | 0.94 | 17.13 | 0.25 | 80K | 0 |
| | Quantile-Mamba | **0.95** | **9.66** | 0.37 | 17K | 0 |
| | Tube-LSTM | 0.94 | 15.56 | 0.28 | 32K | 0 |
| | QD-LSTM | 0.79 | 5.94 | 0.14 | 32K | 0 |
| | DeepAR | 0.90 | 12.79 | 0.11 | 13K | 0 |
| BP | SSVQR | 0.96 | 78.21 | 0 | 325 | 70 |
| | SVQR | **0.95** | **75.73** | **0** | 325 | 0 |
| | Quantile-LSTM | 0.94 | 134.80 | 6.43 | 29K | 0 |
| | Quantile-GRU | 0.94 | 139.88 | 89.54 | 65K | 0 |
| | Quantile-TCN | 0.95 | 110.32 | 14.98 | 120K | 0 |
| | Quantile-Transformer | 0.94 | 124.90 | 18.03 | 80K | 0 |
| | Quantile-Mamba | 0.96 | 108.11 | 161.35 | 30K | 0 |
| | Tube-LSTM | 0.95 | 42.91 | 10.61 | 29K | 0 |
| | QD-LSTM | 0.96 | 159.71 | 60.02 | 29K | 0 |
| | DeepAR | 0.76 | 12.43 | 154.65 | 13K | 0 |

Table 5: Comparison of probabilistic forecasting methods. PICP and MPIW quantify data uncertainty, while $\sigma^2_{opt}$ reflects optimization uncertainty across training runs. Sparsity denotes the percentage of zero coefficients in solution weight. The best model is the one that achieves the lowest MPIW while satisfying the target coverage level of 0.95. FB: Female Births (365×1), MnTemp: Minimum Temperature (3651×1), BP: Beer Production (464×1). #W denotes the number of trainable weights. Tube-LSTM Anand et al. (2026), QD-LSTM Pearce et al. (2018) and DeepAR Salinas et al. (2020) SVQR: Support Vector Quantile Regression, SSVQR: Sparse Support Vector Quantile Regression

formance of deep probabilistic forecasting models that adopt different probabilistic forecasting approaches, including the Tube Loss LSTM (Anand et al. (2026)) and Quality-Driven (QD) Loss LSTM models (Pearce et al. (2018)), as well as the DeepAR model (Salinas et al. (2020)). The QD loss (Pearce et al. (2018)) is the improved version of the LUBE model (Khosravi et al. (2010)) can be used minimized with the gradient descent method in deep learning architecture.

All models were trained to obtain probabilistic forecasts at a target calibration level of $1 - \alpha = 0.95$ and were repeated five times under identical hyper-parameter settings after careful parameter tuning. Quantile-based models estimate the 0.025 and 0.975 quantiles of the predictive distribution.

SVM methods require the tuning of the two parameters namely $C$ and RBF kernel parameter $\gamma$. We have tunned the value of the these parameters using the grid search in the search space $\{2^{-15}, 2^{-14}, ...., 2^{14}, 2^{15}\} \times \{2^{-15}, 2^{-14}, ..., 2^{14}, 2^{15}\}$.

For the probabilistic forecasting tasks, we quantify the model uncertainty using only the *optimization uncertainty* $\sigma_{\text{opt}}^2$, as the estimation of the *input uncertainty* is practically difficult. This is because the sequential nature of the dataset limits the random and independent sampling of training sets.

The deep learning based probabilistic forecasting models such as Quantile-based LSTM, Quantile-based GRU, Quantile-based TCN, Quantile-based Transformers, Quantile-based Mamba, Tube Loss LSTM, Deep AR and QD Loss LSTM models were trained in a Python environment using the Tensor-Flow library. All of them were evaluated on the test set by effectively tuning their parameters on the validation set. A detailed description of tuned hyper-parameters, architectures of the tuned deep neural networks, and probabilistic forecasting results is provided in Appendix A.3 containing the Table 11.

Table 5 presents the numerical results. From a data uncertainty perspective, the SVM-based models achieve PI quality, measured by PICP and MPIW, that is comparable to or better than that of deep learning architectures, despite using significantly fewer trainable weights. Additionally , deep learning models exhibit noticeable *optimization uncertainty*, as indicated by non-zero values of $\sigma_{opt}^2$. Furthermore, SVM-based UQ models are relatively simpler, obtain lower model uncertainty and require the adjustment of significantly fewer parameters compared to complex deep learning models. Overall, these results suggest that the SVM-based approach provides a stable, effective and promising solution for probabilistic forecasting when both data and model uncertainty are considered.

### 5.4.2 Scalability Analysis: SVM versus deep learning models in probabilistic forecasting

To study the characteristics of SVM relative to deep-learning models on a large-scale probabilistic forecasting task, we consider the Amprion energy dataset (Sundblad et al., 2023), which records the wind power generated at 15-minute intervals. We resample it to 30-minute intervals, yielding a processed dataset of first 16K recordings. To assess the effect of dataset size on the learning models, we sample datasets of varying length from the beginning of the series. For example, the Amprion-4K dataset consists of the first 4K wind-power recordings. For each dataset, the last 1,000 points are held out as the test set, and the last 10% of the remaining training points is taken out as the validation set.

We train quantile-based LSTM, GRU, and TCN models and compare their performance against the SVM-based quantile regression model (SVQR). All models target PIs with a nominal coverage of $1 - \alpha = 0.90$. For this purpose, they estimate the $0.05^{th}$ and $0.95^{th}$ quantile functions. We first tune the hyper-parameters of all four models. The tuned parameter values are listed in Table 12 of Appendix A, which also reports the window size $w = 24$ for all considered learning models. This means that the last 24 wind power observations are used to estimate the PI for the wind power recording at the next time step.

The data uncertainty is quantified using the PICP and MPIW metrics. For the model uncertainty, we quantify only $\sigma_{\text{opt}}$, since estimating the *input uncertainty* is practically difficult due to the sequential nature of the time-series dataset. To compute $\sigma_{\text{opt}}$, we train each learning model over $M = 10$ independent trials. Table 6 reports the performance of the different deep-learning architectures against SVM (SVQR), quantifying the data uncertainty and model uncertainty across datasets of varying size.

We note that the PI quality of the SVM model remains comparable to, and sometimes better than, that of deep-learning models. Moreover, the SVM estimates are stable across trials, yielding zero *optimization uncertainty*, whereas the deep-learning models exhibit *optimization uncertainty*.

However, the SVM model seems to be impractical choice with the growth of the size of dataset. Figure 4(a) plots the training time of each learning model against dataset size. The training time of the SVM model grows steeply, by roughly three orders of magnitude over the range considered, whereas the deep-learning models grow far more slowly and remain nearly constant. Consequently, training the SVM model becomes computationally impractical beyond 10K data points. Figure 4(b) plots $\sigma_{\text{opt}}$ against dataset size, showing that the SVM model remains deterministic while the *optimization uncertainty* of the deep-learning models decreases as the dataset grows, making the SVM's determinism advantage less pronounced on large datasets.

| Model | Metric | 4k | 6k | 8k | 10k | 12k | 16k |
|-------|--------|-----|-----|-----|-----|-----|-----|
| SVQR | PICP | 0.890 | 0.863 | 0.861 | 0.924 | 0.952 | 0.899 |
| | MPIW | 23.34 | 30.53 | 23.65 | 26.81 | 15.99 | 14.55 |
| | $\sigma_{\mathrm{opt}}$ | 0 | 0 | 0 | 0 | 0 | 0 |
| | Time (s) | 20.3 | 75.9 | 394.0 | 705.2 | 6,772 | 24,775 |
| LSTM | PICP | 0.930±0.010 | 0.905±0.022 | 0.889±0.014 | 0.930±0.019 | 0.949±0.010 | 0.900±0.042 |
| | MPIW | 24.86±1.24 | 33.17±2.40 | 25.00±1.03 | 27.48±1.27 | 16.06±1.11 | 14.77±1.29 |
| | $\sigma_{\mathrm{opt}}$ | 0.0316 | 0.0458 | 0.0141 | 0.0265 | 0.0200 | 0.0200 |
| | Time (s) | 6.5±0.1 | 13.8±3.8 | 23.1±4.2 | 31.4±0.1 | 38.7±0.6 | 63.5±4.3 |
| GRU | PICP | 0.920±0.026 | 0.918±0.027 | 0.894±0.037 | 0.930±0.021 | 0.932±0.043 | 0.939±0.061 |
| | MPIW | 27.50±2.23 | 40.51±6.75 | 29.68±5.85 | 31.84±4.40 | 18.66±2.14 | 20.92±3.81 |
| | $\sigma_{\mathrm{opt}}$ | 0.1407 | 0.1334 | 0.0686 | 0.0762 | 0.0600 | 0.0671 |
| | Time (s) | 11.9±0.6 | 20.3±1.3 | 33.7±0.4 | 42.7±2.4 | 47.1±4.5 | 66.2±5.7 |
| TCN | PICP | 0.921±0.015 | 0.923±0.030 | 0.906±0.027 | 0.940±0.014 | 0.942±0.018 | 0.930±0.021 |
| | MPIW | 25.57±1.10 | 37.32±4.95 | 26.77±1.87 | 30.82±2.21 | 16.72±1.60 | 16.37±0.81 |
| | $\sigma_{\mathrm{opt}}$ | 0.0927 | 0.0728 | 0.0224 | 0.0346 | 0.0283 | 0.0245 |
| | Time (s) | 21.5±0.1 | 38.5±3.3 | 40.2±10.3 | 37.8±0.3 | 46.1±0.1 | 62.8±0.1 |

Table 6: Comparison of SVM (SVQR) against popular deep-learning models for probabilistic forecasting on the Amprion wind-power dataset (Sundblad et al., 2023), with target coverage $1 - \alpha = 0.90$. All models are trained over 10 trials. The SVM(SVQR) estimates are deterministic and hence do not vary across trials. The tuned hyper-parameters are listed in Table 12 of Appendix A.

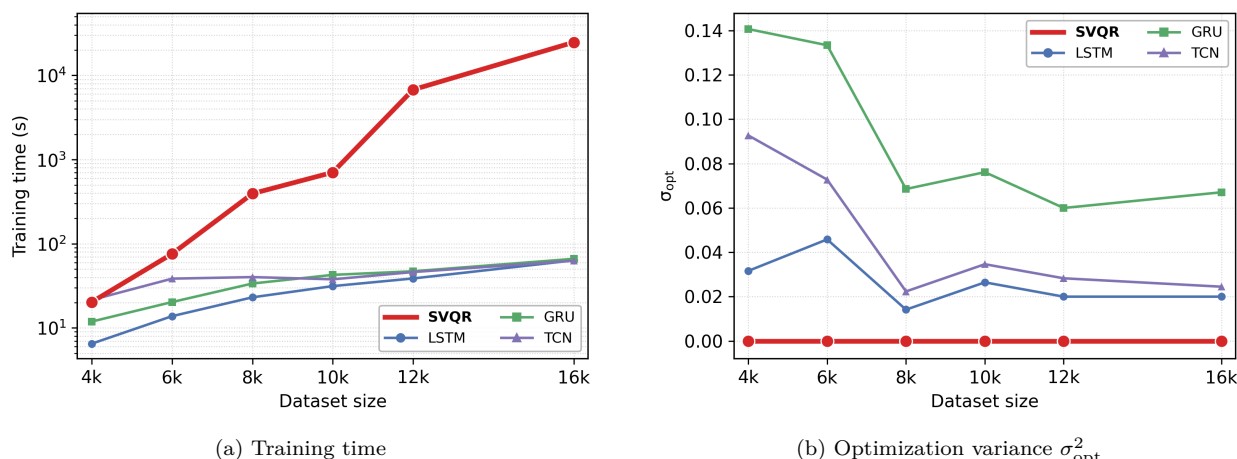

(a) Training time

(b) Optimization variance $\sigma_{\mathrm{opt}}^2$

Figure 4: Scalability trade-off on the Amprion data (Sundblad et al., 2023). (a) SVM (SVQR) training time grows by roughly three orders of magnitude with dataset size (log scale), whereas the neural models remain nearly constant. (b) SVM (SVQR) is deterministic ($\sigma_{\mathrm{opt}} = 0$), but the optimization variance of the neural models also diminishes as the dataset grows, so SVM (SVQR) determinism advantage becomes less pronounced at scale.

### 5.4.3 SVM probabilistic forecasting under distribution shift

Probabilistic forecasting is essentially the task of estimating a PI in an autoregressive setting. However, the PI produced by a probabilistic forecasting model does not come with a coverage guarantee, and this matters because the predictive distribution of real-world time series typically evolves over time; maintaining valid coverage despite such distribution shift can be critical for downstream decision-making. To address this, Gibbs and Candès developed the Adaptive Conformal Inference (ACI) framework in thier work (Gibbs &

Candès (2021)), which provides coverage guarantees even when the underlying data distribution shifts over time. The SVM-based probabilistic forecasting model introduced in this paper can be combined with ACI to obtain the same guarantees: the SVM is used only to produce the base quantile functions, which ACI then calibrates online, without any additional retraining.

To compare the SVM and deep learning models under the ACI setting, we use the same Amprion wind power dataset (Sundblad et al., 2023) employed in our earlier probabilistic forecasting experiments, sampled at 30-minute intervals. The first 5,000 recordings are used to train the base predictors, and the following 10,000 recordings form the test sequence, giving a test-to-train ratio of 2:1. Because wind power generation exhibits strong seasonal and weather-driven non-stationarity, this chronological split induces a genuine distribution shift between the training and test periods. The target coverage $1 - \alpha$ is set to 0.90.

We reuse the hyper-parameters (including the window size), already tuned for the LSTM and SVM models in our earlier probabilistic forecasting experiments (listed in the Table 12 of Appendix A), so as to avoid repeating the tuning process, and retrain both models on the training set. Once trained, the weights of both models are frozen. Each frozen model then produces a base PI $[\hat{f}_{0.05}(x_t), \hat{f}_{0.95}(x_t)]$ for every point in the test sequence with the rolling window approach.

The test sequence is then processed by ACI in an online, step-by-step manner. At each step $t$, ACI maintains a rolling calibration window of the most recent 500 conformity scores, computed with $s_i = \max\left(\hat{f}_{0.05}(x_i) - y_i,\ y_i - \hat{f}_{0.95}(x_i)\right)$ and uses them to compute an adaptive correction $\delta_t$, taken as the $(1 - \alpha_t)(1 + \frac{1}{500})$-th empirical quantile of these scores. The base interval is then widened or tightened using $\delta_t$ to give the adjusted prediction set

$$\hat{C}_t = \left[\hat{f}_{0.05}(x_t) - \delta_t,\ \hat{f}_{0.95}(x_t) + \delta_t\right].$$

After $x_t$ is observed, the miscoverage level is updated for the next step as

$$\alpha_{t+1} = \alpha_t + \gamma\left(\alpha - \mathbf{1}\{y_t \notin \hat{C}_t\}\right), \tag{19}$$

so that $\alpha_t$ decreases (widening subsequent intervals) after a miscoverage event, and increases back toward the nominal level (tightening subsequent intervals) after correct coverage. We tune and use a step size $\gamma = 0.001$ and target miscoverage rate $\alpha = 0.10$ throughout. Crucially, the ACI loop requires no retraining of the underlying model. It works entirely with the base quantile predictions and conformity scores that have already been computed, and simply uses them to widen or shrink the PI at each step. This makes ACI a lightweight calibration layer that can be placed on top of any base model, whether SVM or a deep learning model, at no extra training cost.

| Model | PICP | MPIW | Train Time (s) |
|---|---|---|---|
| SVM | 0.889 | 21.53 | 185.1 |
| SVM+ACI | 0.899 | 24.57 | 185.6 |
| LSTM | 0.948 | 36.23 | 48.8 |
| LSTM+ACI | 0.900 | 32.91 | 49.1 |

Table 7: Comparison of SVM and LSTM models before and after ACI re-calibration.

Table 7 presents the comparison of the SVM and LSTM model performances before and after using the ACI framework. In case of the SVM, the initial estimated PI coverage was a bit short of the target 0.90, which is well improved by the ACI framework. In case of the LSTM, the estimated PI was wider than required, which was also improved by the ACI framework. Further, we note that the overall quality of the PI estimated through the SVM after using the ACI framework is better than that obtained by the LSTM with the ACI framework. This shows that the use of the SVM model within the ACI setting is also effective and promising. However, the overall training time obtained by SVM model is much higher than that of LSTM model.

### 5.4.4   Pretrained zero-shot time-series models: A model uncertainty perspective

Recently, several zero-shot time-series foundation models, such as Chronos (Ansari et al., 2024), TimesFM (Das et al., 2024), and Moirai (Woo et al., 2024), have been proposed to capture general temporal patterns and enable faster deployment with improved generalization across domains. For a given time-series forecasting problem, these models do not require training from scratch and can generate forecasts directly after deployment without task-specific training or fine-tuning. They rely on complex and carefully designed neural architectures and are typically pretrained on large-scale time-series datasets. Since these models are based on neural network architectures, they may still be affected by *input uncertainty* and *optimization uncertainty*. However, quantifying such uncertainty would require retraining the overall model multiple times and examining the variability in the learned weights or predictions, which may not be practical for large-scale foundation models.

We used the Amprion-4K energy dataset from our previous probabilistic forecasting experiment and supplied it to Chronos v1 (Ansari et al., 2024) to generate probabilistic forecasts for the next 1000 test points, following the experimental setup used for the probabilistic forecasting results listed in the Table 6. The forecasts were obtained using a rolling one-step-ahead forecasting approach. Unlike traditional learning models, whose predictions remain deterministic during inference once the model parameters are learned, Chronos v1 may exhibit test-time variability in its estimates. To examine this, we queried Chronos v1 (Ansari et al., 2024) to obtain PI with target coverage $1 - \alpha = 0.90$ over 10 independent trials using the same Amprion-4K energy dataset. We observed variability in the estimated PIs across the trials, which appears to emerge from the stochasticity involved in its generative sampling mechanism and quantify it using

$$\sigma^2_{\text{test}} = \frac{1}{9k} \sum_{i=1}^{k} \sum_{j=1}^{10} \left( f_{0.05}^{(j)}(x_i) - \bar{f}_{0.05}(x_i) \right)^2 + \frac{1}{9k} \sum_{i=1}^{k} \sum_{j=1}^{10} \left( f_{0.95}^{(j)}(x_i) - \bar{f}_{0.95}(x_i) \right)^2 \qquad (20)$$

where $f_q^{(j)}(x_i)$ denotes the $q^{th}$ quantile estimate obtained in the $j^{th}$ test trial, and $\bar{f}_q(x_i)$ denotes the average of the $q^{th}$ quantile estimates obtained over 10 independent trials using the zero-shot pretrained model. In our experiment over 10 independent test trials, we obtain the $\sigma_{\text{test}} = 5.66$.

The average mean PICP obtained by Chronos v1 (Ansari et al., 2024) was 0.82, with a mean MPIW value of 22.64. The mean PICP falls short of the target coverage level of 0.90. The PI quality obtained by Chronos v1 (Ansari et al., 2024) also appears to be poorer than that achieved by the traditional learning models trained from scratch, as reported in Table 6. In Table 6, the SVM (SVQR) model achieves a PICP of 0.89 with an MPIW value of 23.34 upon the Amprion-4K dataset, whereas LSTM, GRU, and TCN achieve PICP values greater than 0.92, with MPIW values in the range of 24.50–27.50.

However, the important and notable observation is the variance ($\sigma_{\text{test}} = 5.66$.) in the estimated PIs produced by Chronos v1 (Ansari et al., 2024) across different test trails. In contrast, this variance remains zero during inference for traditional learning models, as they become fully deterministic once training is completed.

## 6   Conclusions and Implications

From a model uncertainty perspective, the SVM model often yields overall lower model uncertainty, comprising both *input uncertainty* and *optimization uncertainty*, than its counterpart learning models. Likewise, from a data uncertainty perspective, it produces PIs of comparable and good quality relative to those models. This advantage stems from SVM's simple kernel based architecture combined with a convex loss formulation, which guarantees a more stable and globally optimal solution while retaining the capacity to model complex relationships. These properties make SVM a promising candidate for UQ modeling, particularly on small and moderate scale datasets. As the dataset size grows, however, these advantages appear to diminish. Moreover, the scalability of SVM models for large scale and online data processing remains a bottleneck and a challenging open problem for SVM researchers.

In view of above facts, we encourage the adoption of SVM-based UQ methods across a wide range of application domains, in small- and moderate-scale data regimes where stability and reliability are critical.

SVMs provide stable, optimal, and reproducible solutions, making their UQ estimates especially valuable in medical, biomedical, and clinical applications, where datasets are often limited, feature dimensionality is high, and trustworthy uncertainty estimates are more important than marginal gains in point prediction accuracy.

In UQ regression tasks, particularly for high-dimensional tabular data, careful attention to feature selection is essential, as it not only reduces model complexity and improves interpretability but also enhances the quality of PIs by mitigating spurious sources of uncertainty. The design of the stable feature selection algorithm for the PI estimation in non-linear case requires the future attention of UQ researchers.

While deep learning models can be effective for probabilistic forecasting on large-scale datasets, their deployment in data-limited settings often leads to over-parameterization, heightened model uncertainty, and unstable solutions. In such cases, we recommend evaluating the solution quality of SVM-based models before training large and complex deep architectures.

To provide a clearer road-map for UQ researchers interested in SVM-based methods, Appendix B presents a detailed comparative analysis of existing SVM-based UQ methods, along with their strengths and limitations.

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

# A    Additional numerical results and experimental details

## A.1    Parameter tuning and ablation study for the experiments in Section 5.1

We carefully tuned the hyper-parameters of all participating models for the experiments conducted on the artificial dataset in Section 5.1. The selected hyper-parameter values for each learning model are reported in Table 8. All hyper-parameters were tuned using a grid search procedure, where an appropriate search grid was defined for each hyper-parameter.

For the NN model, we employed a single hidden layer to mitigate overfitting. The number of neurons in the hidden layer was selected from the set $2^i : i = 4, 5, \ldots, 10$ using the grid search method. The ablation study presented in Table 9 shows that using 512 neurons yields the best-quality PI while maintaining a manageable level of *optimization uncertainty* ($\sigma_{opt}$). Furthermore, the batch size and learning rate were chosen from the search spaces $\{2^i : i = 3, 4, \ldots, 8\}$ and $\{0.001, 0.002, \ldots, 0.3\}$, respectively.

For the SVM model, only two hyper-parameters need to be tuned: the RBF kernel parameter $\gamma$ and the regularization parameter $C$, making the hyper-parameter tuning process considerably simpler than that of the other models. Both hyper-parameters were optimized using a grid search over the set of grid points $\{2^i : i = -12, -11.5, 11, \ldots, 12\}$.

For the QRF model, the number of trees was selected from the set $\{50, 100, 150, 200, 300, 500\}$ using the grid search method. In addition, the minimum number of samples per leaf was selected from the set $\{5, 10, 15, 20, 30, 50\}$.

For the XGBoost model, the number of estimators was selected from the set $\{50, 100, 200, 300, 500\}$, while the maximum tree depth (max_depth) was selected from the set $\{1, 2, 3, 4, 5, 7, 10\}$. The subsampling ratio (subsample) was selected from the set $\{0.4, 0.6, 0.8, 1.0\}$. Figure 5 presents the ablation study by plotting the Mean of RMSE obtained from the pair of estimated quantile functions against the number of estimators and the maximum tree depth, thereby justifying the corresponding choices reported in Table 8. Although setting subsample = 1.0 makes the XGBoost estimates deterministic, a subsampling ratio of 0.8 yielded the best PI quality in our experiments.

| Model | hyper-parameter | Value |
|-------|-----------------|-------|
| **NN** | Hidden layer neurons | 512 |
| | Epochs | 100 |
| | Batch Size | 32 |
| | Learning Rate | 0.003 |
| **QRF** | $N_{estimators}$ | 50 |
| | Min Samples Leaf | 5 |
| | Max Features | sqrt |
| **SVM** | Kernel | RBF |
| | Gamma ($\gamma$) | $2^{-0.5} \approx 0.707$ |
| | Regularization ($C$) | $2^{3.5} \approx 11.31$ |
| **XGBoost** | $N_{estimators}$ | 200 |
| | Max Depth | 2 |
| | Learning Rate | 0.05 |
| | Subsample | 0.8 |
| | Col sample by Tree | 0.8 |
| | Min Child Weight | 3 |
| | Regularization ($\lambda$) | 2.0 |
| **GPR** | Kernel | RBF |
| | Kernel length Scale | 0.707 |
| | Noise Variance ($\alpha$) | 0.5 |
| **NGB** | Base Learner | Decision Tree |
| | $N_{estimators}$ | 300 |
| | Learning Rate | 0.01 |
| | Max Depth | 2 |
| | Min Samples Leaf | 10 |
| | Minibatch Fraction | 1.0 |

Table 8: Tuned hyper-parameter configurations for each used learning models estimated on artificial dataset at Section 5.1

| Nodes $h$ | Sum of RMSEs | | $\sigma_{\text{opt}}$ | |
|---|---|---|---|---|
| | Val | Test | Val | Test |
| 16 | 0.500 | 0.509 | 0.223 | 0.221 |
| 32 | 0.384 | 0.392 | 0.250 | 0.258 |
| 64 | 0.263 | 0.266 | 0.131 | 0.134 |
| 128 | 0.217 | 0.224 | 0.087 | 0.084 |
| 256 | 0.154 | 0.161 | 0.080 | 0.080 |
| **512** | **0.109** | **0.113** | 0.160 | 0.155 |
| 1024 | 0.110 | 0.107 | 0.250 | 0.236 |

Table 9: An ablation study on the number of neurons (h) in the hidden layer was conducted on the artificial dataset generated in the experiments of Section 5.1. The results show that 512 neurons is a good choice for this parameter on the artificial dataset.

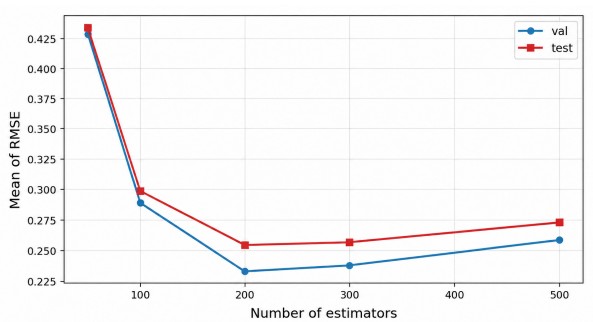 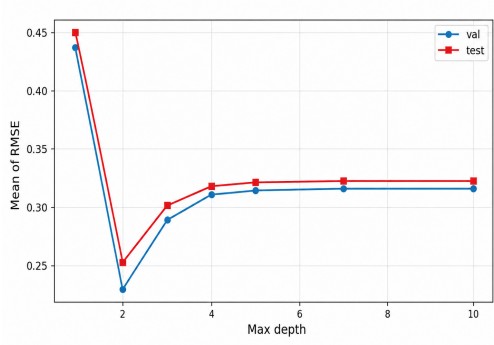

Figure 5: XGBoost ablation over number of estimators, $n_{\text{est}} \in \{50, 100, 200, 300, 500\}$ and max depth $\in \{1, 2, \dots 10\}$ for artificial dataset generated in Section 5.1 for a fixed seed. The number of estimators was fixed to 200 with max depth of 2 for the experiment with XGBoost in Section 5.1.

## A.2 Parameter tuning details for experiments in Section 5.3

Before obtaining the numerical results listed in Table 4, we carefully tuned the hyper-parameters of each participating learning model to obtain the best PI estimates without touching the test set. We performed a grid search to select the optimal hyper-parameters for the NN, SVM, QRF, and XGBoost models. The corresponding search space is the same as that detailed in Appendix A.1. The tuned hyper-parameters for each learning model evaluated in Table 4 are listed in Table 10.

## A.3 Details about the probabilistic forecasting experiments of Section 5

Datasets:- We consider three popular time-series datasets namely Female Births ($365 \times 1$) (Datamarket.com), Minimum Temperature ($3651 \times 1$) (machinelearningmastery.com) and Beer Production ($464 \times 1$) (Australian (1996)). We have used the 70% of dataset as training set and rest of them are used for testing. Out of the training set, the last 10% of the observations have been used for the validation set.

Baseline Methods:- We also train several recent and widely adopted Quantile based deep learning architectures for probabilistic forecasting developed in distribution-free setting, including Quantile-based LSTM, Quantile-based GRU, Quantile-based TCN and Quantile-based Transformers. The SVQR and SSVQR models are fundamentally quantile-based probabilistic forecasting approaches; therefore, when comparing them with deep learning counterparts, we should only consider quantile-based deep learning methods. However, for the sake of extensive comparisons, we also report the performance of deep probabilistic forecasting models that adopt different probabilistic forecasting approaches, including the Tube Loss LSTM and Quality-Driven

| Model | Hyper-parameter | Boston | Concrete | Servo | AutoMPG | RealEstate |
|---|---|---|---|---|---|---|
| CQR-NN | hidden_dim | 512 | 128 | 32 | 512 | 64 |
| | epochs | 300 | 300 | 100 | 200 | 200 |
| | learning_rate | 0.05 | 0.05 | 0.05 | 0.05 | 0.05 |
| | batch_size | 32 | 128 | 128 | 128 | 256 |
| SVQR+CP | kernel parameter $\gamma$ | 0.0625 | 0.0125 | 0.0625 | 0.0250 | 0.0233 |
| | $C$ | 16 | 4096 | 256 | 512 | 32 |
| QRF+CP | n_estimators | 50 | 150 | 200 | 50 | 200 |
| | min_samples_leaf | 5 | 1 | 1 | 1 | 1 |
| XGBoost+CP | n_estimators | 500 | 500 | 200 | 500 | 200 |
| | max_depth | 8 | 3 | 8 | 5 | 5 |
| | learning_rate | 0.05 | 0.1 | 0.01 | 0.05 | 0.1 |
| | subsample | 1 | 1 | 1 | 1 | 1 |

Table 10: Tuned hyper-parameters per dataset for CQR-NN, SVQR+CP, QRF+CP and XGBoost+CP. For SVM (SVQR), $\gamma$ is the RBF kernel parameter (kernel: $\exp(-\gamma\|x - x'\|^2)$). XGBoost is trained with `subsample`$= 1.0$ (fixed, deterministic).

(QD) Loss LSTM models (Pearce et al. (2018)), as well as the DeepAR model (Salinas et al. (2020)). The QD loss (Pearce et al. (2018)) is the improved version of the LUBE model which can be used minimized with the gradient descent method in deep learning architecture.

Experimental Setup:- The SSVQR, SVQR and LS-SVR model based probabilistic forecasting model were trained in MATLAB. All of the models were asked to obtain the probabilistic forecast for target calibration $1 - \alpha = 0.95$. For this, they estimate the $0.975^{th}$ and $0.025^{th}$ quantiles of the predictive distribution. All of them requires the tuning of the two parameters namely $C$ and RBF kernel parameter $\gamma$. We have tunned the value of the these parameters using the grid search in the search space $\{2^{-15}, 2^{-14}, ...., 2^{14}, 2^{15}\} \times \{2^{-15}, 2^{-14}, ..., 2^{14}, 2^{15}\}$.

The deep learning based probabilistic forecasting models such as Quantile-based LSTM, Quantile-based GRU, Quantile-based TCN, Quantile-based Transformers, Tube Loss LSTM, Deep AR and QD Loss LSTM models were trained in a Python environment using the Tensor-Flow library. All of them were evaluated on the test set by effectively tuning their parameters on the validation set.

| Dataset | Model | Architecture | Number of weights | Window Size |
|---------|-------|--------------|-------------------|-------------|
|  | SVQR | Kernel | 256 | 10 |
|  | SSVQR | Kernel | 256 | 15 |
|  | LS-SVR | Kernel | 256 | 20 |
|  | Quantile LSTM | LSTM [100] | 30 K | 25 |
|  | Tube LSTM | LSTM [100] | 30 K | 25 |
| **Female Births** | QD LSTM | LSTM [100] | 30 K | 25 |
|  | DeepAR | LSTM [40,40] | 13 K | 28 |
|  | Quantile GRU | GRU [128] | 65 K | 12 |
|  | Quantile TCN | TCN [32,64,64] | 120 K | 12 |
|  | Quantile Transformer | Transformer [64,4,2] | 80 K | 12 |
|  | SVQR | Kernel | 2 556 | 12 |
|  | SSVQR | Kernel | 2 556 | 18 |
|  | LS-SVR | Kernel | 2 556 | 22 |
|  | Quantile LSTM | LSTM [16,8] | 32 K | 28 |
|  | Tube LSTM | LSTM [16,8] | 32 K | 28 |
| **Minimum Temp.** | QD LSTM | LSTM [16,8] | 32 K | 28 |
|  | DeepAR | LSTM [40,40] | 13 K | 56 |
|  | Quantile GRU | GRU [128] | 65 K | 12 |
|  | Quantile TCN | TCN [32,64,64] | 120 K | 12 |
|  | Quantile Transformer | Transformer [64,4,2] | 80 K | 12 |
|  | SVQR | Kernel | 325 | 8 |
|  | SSVQR | Kernel | 325 | 14 |
|  | LS-SVR | Kernel | 325 | 18 |
|  | Quantile LSTM | LSTM [64,32] | 29 K | 24 |
|  | Tube LSTM | LSTM [64,32] | 29 K | 24 |
| **Beer Prod.** | QD LSTM | LSTM [64,32] | 29 K | 24 |
|  | DeepAR | LSTM [40,40] | 13 K | 48 |
|  | QuantileGRU | GRU [128] | 65 K | 12 |
|  | Quantile TCN | TCN [32,64,64] | 120 K | 12 |
|  | Quantile Transformer | Transformer [64,4,2] | 80 K | 12 |

Table 11: Comparison of the complexity of used SVM and deep learning based probabilistic forecasting models on benchmark datasets. LSTM [16,8] means that the used LSTM model has two hidden layer containing 16 and 8 neurons respectively. While the SVM-based model delivers competitive, stable performance with zero model uncertainty only with a few hundred parameters, deep learning models require optimizing thousands of parameters.

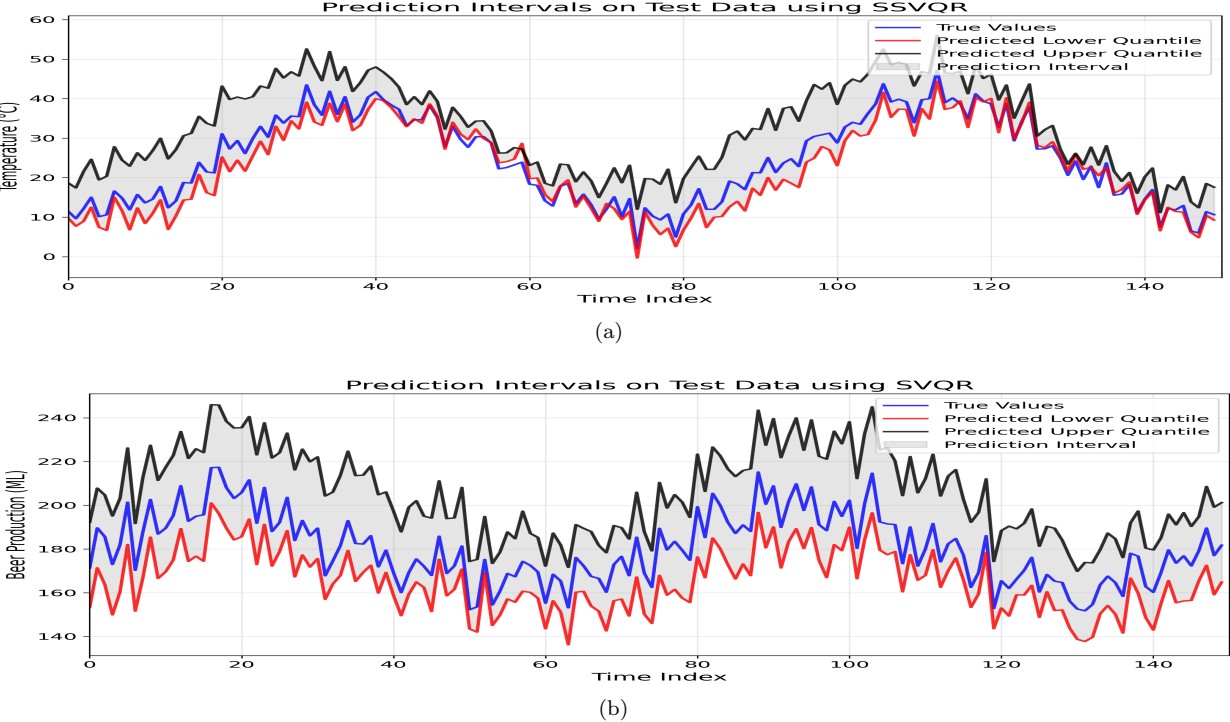

Figure 6: Probabilistic forecasting with SSVQR model on daily (a) Temperature and (b) Beer Production dataset

|  |  | SVQR | LSTM | GRU | TCN |
|---|---|---|---|---|---|
| *Shared* | Window size $w$ | 24 | 24 | 24 | 24 |
|  | Quantile pairs | $(0.05, 0.95)$ | $(0.05, 0.95)$ | $(0.05, 0.95)$ | $(0.05, 0.95)$ |
|  | Target coverage | 0.90 | 0.90 | 0.90 | 0.90 |
|  | Data-size | | {4k, 6k, 8k, 10k, 12k, 16k} | | |
| *SVQR* *(RBF kernel)* | Kernel | RBF: $\exp(-\gamma\|x-x'\|^2)$ | — | — | — |
|  | $\gamma$ | $2^{20} \approx 1.05\times10^6$ | — | — | — |
|  | $C$ | $2^{16} = 65\,536$ | — | — | — |
| *NN* *(exp6)* | Hidden size | — | 128 | 128 | 256 |
|  | Learning rate | — | $10^{-2}$ | $10^{-2}$ | $10^{-2}$ |
|  | Batch size | — | 32 | 64 | 128 |
|  | Epochs | — | 50 | 50 | 50 |
|  | Optimiser / scheduler | — | Adam + `ReduceLROnPlateau` (patience 5, factor 0.5) | | |

Table 12: Hyper-parameters tuned for Amprion energy dataset (He et al., 2019).

## A.4 Effect of Test Set Size on Model and Data Uncertainty Estimates

We conduct experiments to examine the impact of dataset size upon the *data uncertainty* and the *model uncertainty*. To this end, we repeat the experiment of Section 5.1 with the test set size varied over $\{600, 1200, 1800, 2400, 3000\}$, while keeping the training sets, hyper-parameter settings, and experimental setup fixed. The training set size $m$ is fixed to 100. Table 13 reports the numerical results. The total model uncertainty estimates, comprising both *input uncertainty* and *optimization uncertainty*, together with the data uncertainty estimates measured through PI quality by the Mean of RMSEs in (18), are plotted against the test set size in Figure 7. Further, the *input uncertainty* and *optimization uncertainty* are plotted separately against the test set size in Figure 8.

We observe that both the model uncertainty and data uncertainty estimates do not depend strongly upon the test set size. In contrast, the experiments in Section 5.1 show that these estimates depend heavily on the training set size. Moreover, SVM consistently achieves the lowest RMSE and $\sigma^2_{\text{model}}$ across all test set sizes.

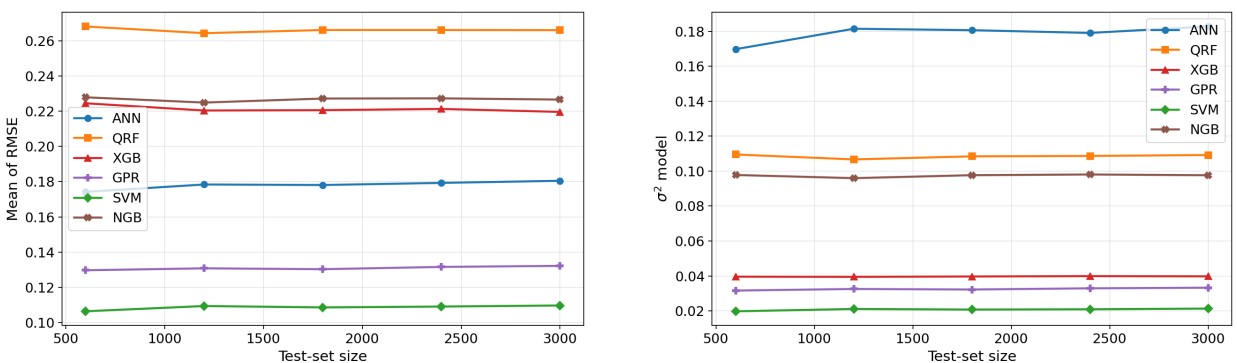

Figure 7: **Left:** Data uncertainty estimates (PI quality), measured by the Mean of RMSEs in (18), against varying test set size for a fixed training set of size 100. **Right:** Total model uncertainty estimates, $\sigma^2_{\text{model}} = \sigma^2_{\text{opt}} + \sigma^2_{\text{in}}$, against test set size. The model and data uncertainty estimates do not depend significantly on the test set size. Moreover, SVM consistently achieves the lowest RMSE and $\sigma^2_{\text{model}}$ across all test set sizes.

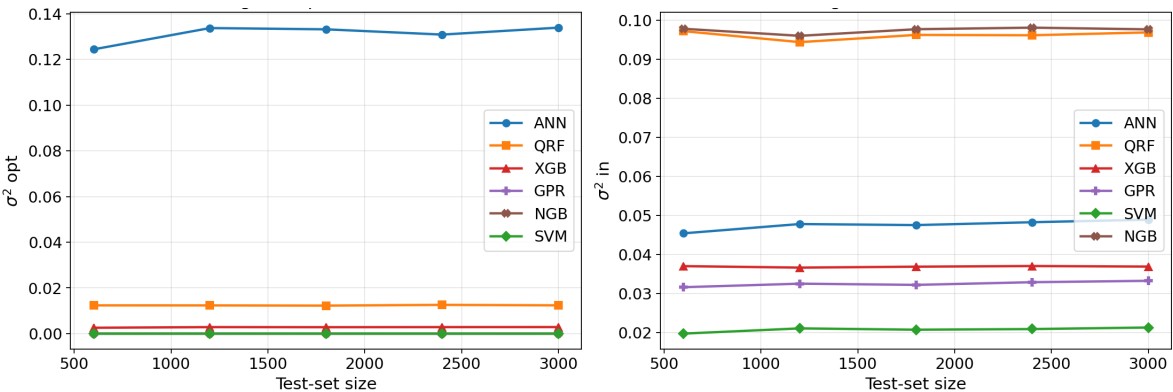

Figure 8: **Left:** Plot of *optimization uncertainty* against different test-set sizes. **Right:** Plot of *input uncertainty* against different test-set sizes. The training-set size was fixed at 100.

| Model | Test size | Mean RMSE | $\sigma^2_{\text{opt}}$ | $\sigma^2_{\text{in}}$ | $\sigma^2_{\text{model}}$ |
|---|---|---|---|---|---|
| ANN | 600 | 0.1742 | 0.1244 | 0.0454 | 0.1698 |
| | 1200 | 0.1784 | 0.1337 | 0.0478 | 0.1815 |
| | 1800 | 0.1781 | 0.1332 | 0.0475 | 0.1807 |
| | 2400 | 0.1793 | 0.1309 | 0.0483 | 0.1791 |
| | 3000 | 0.1805 | 0.1339 | 0.0489 | 0.1828 |
| QRF | 600 | 0.2681 | 0.0124 | 0.0972 | 0.1095 |
| | 1200 | 0.2643 | 0.0123 | 0.0944 | 0.1067 |
| | 1800 | 0.2661 | 0.0122 | 0.0962 | 0.1085 |
| | 2400 | 0.2661 | 0.0126 | 0.0961 | 0.1087 |
| | 3000 | 0.2660 | 0.0124 | 0.0969 | 0.1092 |
| SVM | 600 | 0.1064 | 0.0000 | 0.0197 | 0.0197 |
| | 1200 | 0.1094 | 0.0000 | 0.0211 | 0.0211 |
| | 1800 | 0.1086 | 0.0000 | 0.0207 | 0.0207 |
| | 2400 | 0.1091 | 0.0000 | 0.0209 | 0.0209 |
| | 3000 | 0.1097 | 0.0000 | 0.0213 | 0.0213 |
| XGBoost | 600 | 0.2245 | 0.0026 | 0.0370 | 0.0396 |
| | 1200 | 0.2204 | 0.0028 | 0.0366 | 0.0395 |
| | 1800 | 0.2206 | 0.0028 | 0.0369 | 0.0397 |
| | 2400 | 0.2213 | 0.0028 | 0.0370 | 0.0399 |
| | 3000 | 0.2196 | 0.0029 | 0.0369 | 0.0398 |
| GPR | 600 | 0.1297 | 0.0000 | 0.0316 | 0.0316 |
| | 1200 | 0.1308 | 0.0000 | 0.0325 | 0.0325 |
| | 1800 | 0.1303 | 0.0000 | 0.0322 | 0.0322 |
| | 2400 | 0.1316 | 0.0000 | 0.0329 | 0.0329 |
| | 3000 | 0.1322 | 0.0000 | 0.0333 | 0.0333 |
| NGB | 600 | 0.2279 | 0.0000 | 0.0978 | 0.0978 |
| | 1200 | 0.2249 | 0.0000 | 0.0960 | 0.0960 |
| | 1800 | 0.2272 | 0.0000 | 0.0977 | 0.0977 |
| | 2400 | 0.2273 | 0.0000 | 0.0981 | 0.0981 |
| | 3000 | 0.2266 | 0.0000 | 0.0976 | 0.0976 |

Table 13: Study of the impact of the test set size upon the data and model uncertainty estimates. Experiments performed in Section 5.1 was repeated with varying test set size with fixed training set of size $m = 100$.

## B  Comparative analysis of PI estimation method in SVM

In this section, we present a comparative analysis of all prediction interval (PI) estimation methods proposed in the SVM literature.

### B.1 Least Squares Support Vector Regression

For estimating the mean regression using training set $T$, the LS-SVR model Suykens et al. (2002) minimizes the least square loss function along with the $L_2$-norm of regularization in the following problem.

$$\min_{(w,b,\xi)} \frac{1}{2} w^T w + C \sum_{i=1}^{m} (\xi_i^2)$$

subject to,

$$y_i - (w^T \phi(x) + b) = \xi_i, \ i = 1, 2, ..m. \tag{21}$$

The solution of problem (21) can be obtained through the following system of equations

$$\begin{bmatrix} 0 & e^T \\ e & K(A, A^T) + \frac{2}{C}I \end{bmatrix} \begin{bmatrix} b \\ \alpha \end{bmatrix} = \begin{bmatrix} 0 \\ Y \end{bmatrix}, \tag{22}$$

where $K(A, A^T)$ is an $m \times m$ kernel matrix constructed from the training set $T$, $e$ is an $m$-dimensional column vector of ones, and $I$ represents the $m \times m$ identity matrix. After obtaining the $(b, \alpha)$ from (22), the LS-SVR estimate the regression function for a given $x \in \mathbb{R}^n$ using

$$f(x) = \sum_{i=1}^{m} k(x_i, x)\alpha_i + b. \tag{23}$$

### B.2 PI estimation through LS-SVR

One of the naive PI estimation methods in SVM literature follows the normal assumption regarding the distribution of $Y|X$ and estimates its mean through (23) by training the LS-SVR model. The error distribution $\epsilon_i = y_i - f(x_i)$ follows a normal distribution with zero mean and variance $\sigma$. This variance can be estimated from the error computed on the training set $T$. The pair of quantile bounds required for PI is estimated as $(f(\hat{x}) + \epsilon_{\frac{\alpha}{2}}, f(\hat{x}) + \epsilon_{1-\frac{\alpha}{2}})$, where $\epsilon_q$ is the $q^{th}$ quantile of the error. A more refined and bias-corrected PI based on the LS-SVR model was proposed in (De Brabanter et al. (2010); Cheng et al. (2014)).

### B.3 PI estimation through Tube loss in SVM

Anand et al. (2026) have developed the Tube loss for PI estimation and probabilistic forecasting. It can be minimized directly to obtain the bounds of the PI simultaneously. The minimizer of the Tube loss function guarantees the target coverage $1 - \alpha$ asymptotically. Also, the PI tube can also be shifted up or down by tuning its parameter $r$ so that it passes through the denser region of data cloud for minimal PI width. Furthermore, the width of the PI tube can be explicitly minimized in its optimization problem through the parameter $\delta$. It helps to improve the PI width, when the PI tube achieves a coverage higher than the target on the validation set.

The Tube loss function is a kind of two-dimensional extension of the pinball loss function (Koenker & Bassett Jr (1978)). For a given $1 - \alpha \in (0, 1)$ and $u_2 \le u_1$, the Tube loss function is given by

$$\rho_{1-\alpha}^r(u_2, u_1) = \begin{cases} (1-\alpha)u_2, & \text{if} \quad u_2 > 0, \\ -\alpha u_2, & \text{if } u_2 \le 0, u_1 \ge 0 \text{ and } ru_2 + (1-r)u_1 \ge 0, \\ \alpha u_1, & \text{if } u_2 \le 0, u_1 \ge 0 \text{ and } ru_2 + (1-r)u_1 < 0, \\ -(1-\alpha)u_1, & \text{if } u_1 < 0, \end{cases} \tag{24}$$

where $0 < r < 1$ is a user-defined parameter and $(u_2, u_1)$ are errors, representing the deviations of $y$ values from the bounds of PI.

Figure 9 illustrates the Tube loss for $(1 - \alpha) = 0.9$ with $r = 0.5$. For $r = 0.5$, the Tube loss function is a continuous loss function of $u_1$ and $u_2$, symmetrically positioned around the line $u_1 + u_2 = 0$. In all

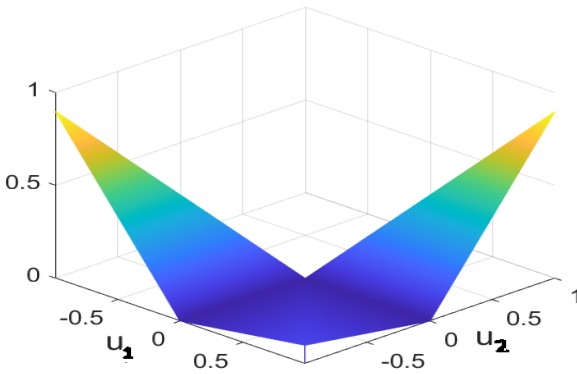

Figure 9: Tube loss function for $1 - \alpha = 0.9$.

experiments with a symmetric noise distribution, the $r$ parameter in the Tube loss function should be set to 0.5 to capture the denser region of $y$ values.

The Tube loss SVM model seeks a pair of kernel-generated functions

$$\mu_1(x) = \sum_{i=1}^{m} k(x_i, x)\alpha_i + b_1 \ \text{ and } \ \mu_2(x) = \sum_{i=1}^{m} k(x_i, x)\beta_i + b_2 \tag{25}$$

by minimizing the optimization problem

$$\min_{(\alpha, \beta, b_1, b_2)} J(\alpha, \beta, b_1, b_2) = \frac{\lambda}{2}(\alpha^T \alpha + \beta^T \beta) + \sum_{i=1}^{m} \rho_{1-\alpha}^r \big( y_i, \big( K(A^T, x_i)\alpha + b_1 \big), \big( K(A^T, x_i)\beta + b_2 \big) \big)$$

$$+ \delta \sum_{i=1}^{m} \big| (K(A^T, x_i)(\alpha - \beta) + (b_1 - b_2) \big|, \tag{26}$$

where $\delta, r$ and $\lambda$ are user-defined parameters and $A$ is the $m \times n$ data matrix containing the training set. Further, details on the Tube SVM problem and its minimization using the gradient descent method can be found in (Anand et al. (2026)).

## B.4 PI estimation through LUBE loss in SVM

The LUBE method (Khosravi et al. (2010)) was originally developed in the NN framework but, was extended to the SVM framework later in (Shrivastava et al. (2014; 2015)) for probabilistic forecasting of electric prices. For the given target confidence $(1 - \alpha)$, and training set $T$, the LUBE SVM model seeks a pair of kernel generated functions of (25), $\mu_1(x)$ and $\mu_2(x)$, which are obtained by minimizing the following loss function

$$CWC = \frac{1}{R} MPIW \big( 1 + \gamma \, (PICP) e^{-\eta(PICP - (1-\alpha))} \big). \tag{27}$$

Here, MPIW is the average width of estimated PI on training set, computed by $\frac{1}{m} \sum_{i=1}^{m} (\mu_2(x_i) - \mu_1(x_i))$. As discussed earlier, PICP is the coverage of the estimated PI and computed using the $PICP = \frac{1}{m} \sum_{i=1}^{m} k_i$, where

$$k_i = \begin{cases} 1, & \text{if } y_i \in [\mu_1(x), \mu_2(x)]. \\ 0, & \text{Otherwise.} \end{cases}$$

Further, the $\gamma(PICP) = \begin{cases} 0, & \text{if } PICP \geq 1 - \alpha, \\ 1, & \text{otherwise,} \end{cases}$, $R$ is the range of response values $y_i$ and $\eta$ is the user-defined parameter.

|  | LS-SVR PI | SVQR PI | LUBE | Tube loss | Sparse SVQR PI |
|---|---|---|---|---|---|
| Distribution-free method | No | Yes | Yes | Yes | Yes |
| Asymptotic guarantees | Only with normal noise | Yes | No | Yes | Yes |
| Direct PI estimation | Yes | No | Yes | Yes | No |
| PI tube movement | No | Yes | No | Yes | Yes |
| Global optimal solution | Yes | Yes | No | No | Yes |
| Re-calibration | No | No | Yes | Yes | No |
| Sparsity | No | No | No | No | Yes |

Table 14: Comparisons of the different PI estimation models available in SVM framework

The major problem with the LUBE cost function (27) is that it is very difficult to be optimized because, the PICP is a step function. Khosravi et al. have solved the LUBE cost function (27) using the Particle Swarm Optimization (PSO) (Kennedy & Eberhart (1995)) to estimate the PI. However, due to sub-optimal solution, high-quality PI is not always observed. Pearce et al. refined the LUBE cost function and use a sigmoidal function to approximate the PICP, allowing the application of the gradient descent method for training the NN for PI estimating in their work (Pearce et al. (2018)).

## B.5 Comparison of PI estimation SVM models

In Table 14, we visualize the desirable properties for a PI estimation model and compare the SVM methods in light of them with a detailed discussion as follows.

(a) Distribution-free method : - The LS-SVR PI model assumes that the underlying noise distribution of the data is normal and may obtain poor estimates otherwise. In the literature, distribution-based PI methods often struggle to achieve consistent performance across various datasets. The SVQR, LUBE, and Tube loss methods provide PI estimation without assuming any specific distribution, allowing them to generate high-quality PI even in the presence of non-normal noise.

(b) Asymptotic guarantees:- A fundamental requirement in PI estimation models is that the obtained PI should guarantee the target coverage $1 - \alpha$ at least asymptotically, which remain missing in the LUBE method. The LS-SVR PI model provides this guarantee only in presence of normal noise. The SVQR and Tube loss based PI methods provide this asymptotic guarantee. The asymptotic guarantee of the SVQR based PI model is implied from the asymptotic guarantee of the pinball loss function minimizer for quantile regression detailed in (Takeuchi et al. (2006)).

(c) Direct PI estimation:- As detailed in Algorithm 1, the SVQR PI model obtains the two bounds of PI by solving pair of SVQR problems one by one. The LUBE and Tube loss-based PI models simultaneously obtain the bounds of the PI by solving a single optimization problem.

(d) PI tube movement:- The PI tube movement is one another important desirable feature for PI estimation. This movement allows the PI to pass through the denser regions of the data cloud, helping to minimize the width of the PI, while achieving the target coverage. The centered PI is ideal only in the presence of the symmetric noise. However, in the presence of asymmetric noise in the data, the width of the prediction interval (PI) can be reduced by shifting it upward or downward without compromising its coverage. The SVQR and Tube loss-based PI models enable PI movement through their parameter $\hat{q}$ and $r$ respectively.

(e) Global Optimal Solution:- One of the attractive features of the standard SVM methods is that they guarantee a global optimal solution by solving a convex optimization problem. However, in PI estimation, only the SVQR and LS-SVR based PI models maintain this guarantee by minimizing the convex loss function in its optimization problems. The Tube loss problem (26) is non-convex and hence fails to guarantee the global optimal solution. Furthermore, the LUBE problem (27) is highly discontinuous and relies on meta-heuristic algorithms for its solution. It often makes the LUBE solution suboptimal, resulting in poor PI quality.

(f) Re-calibration:- As detailed in Algorithm 1, the SVQR PI model obtains the bound of PI by solving the pair of SVQR problems one-by one. It can not explicitly minimize the width of PI in its optimization problem. This limitation prevents SVQR PI from using the recalibration feature. In recalibration, PI models are retrained to reduce interval width, when empirical coverage obtained on validation set exceeds the target $1 - \alpha$ significantly. During retraining, the PI models increase the value of the parameter ($\delta$ in case of the Tube loss) that trade-off the width of the PI against the coverage in the optimization problem. The LUBE and Tube loss-based models explicitly incorporate the minimization of prediction interval (PI) width in their problems, thereby enabling recalibration, which is practically useful for further reducing the PI width.

(g) Sparsity:- Sparsity is yet another promising feature offered by the initial SVM models developed for the classification and regression. However, it remains missing with all PI estimation models developed in the SVM framework.

Only few of components of a sparse vector are non-zero. A sparse solution in a learning model is desirable and important as it offers significant advantages. In case of the linear model, it can help us to drop irrelevant features making the learning task more simple and effective. Further, in case of learning non-linear function also, it reduces the complexity of the model without compromising its predictive ability on train set that makes the model own better generalization ability on unseen test points.

In the SVM literature, two primary approaches are commonly used to obtain sparse solutions, which often lead to confusion. The first approach minimizes the empirical risk with an additional $L_1$-norm regularization term, while the second relies on the $\epsilon$-insensitive loss function to minimize empirical error. However, when estimating a linear regression model of the form $w^T x + b$, with $w \in \mathbb{R}^n$, the latter approach can only induce sparsity in the dual variables (Lagrangian multipliers) and does not guarantee sparsity in $w$. As a result, it cannot eliminate irrelevant features during the learning process. In contrast, explicitly minimizing the $L_1$-norm of $w \in \mathbb{R}^n$ within the optimization problem directly enforces sparsity in $w$, thereby enabling feature selection.

## B.6    Comparing the SVM based PI estimation methods through numerical experiments

We shall perform experiments to compare the SVM based (SVQR), Sparse SVM (SSVQR) PI and LS-SVR based PI estimation model on artificial and benchmark datasets.

**Evaluation metrics and criteria**

Now, we describe in detail the evaluation criteria that will be used for our experiments. In all our experiments, the objective is to estimate PI with a confidence level of either $1 - \alpha = 0.95$ or $1 - \alpha = 0.90$. To construct a PI with $1 - \alpha$ confidence level, we seek the pair of quantile functions $(f_q(x), f_{1+q-\alpha}(x))$, where $0 \leq q \leq 0.05$.

In case of artificial datasets, we know the noise distribution and the true quantile function can be easily computed by the inverse of the cumulative distribution function. Therefore, the quality of the quantile function can be accurately assessed by computing the RMSE between the true and estimated quantile functions.

However, to estimate the quality of the quantile function in the absence of information about the noise distribution, we can use the Coverage Probability (CP). For given a test set $\{(x_i, y_i) : x_i \in \mathbf{R}^n, y \in \mathbf{R}, i = 1, 2, ....k\}$ and estimate of the $q^{th}$ quantile function $f_q(x)$, the CP is computed by $\frac{|\{y_i : y_i \leq f_q(x_i)\}|}{k}$. It measures the fraction of $y$ values falling below the estimated quantile function.

For evaluating the overall quality of PI estimation, we use PICP and MPIW as assessment criteria. If the $[f_q(x), f_{1+q-\alpha}(x)]$ is the estimate of the PI, then the PICP on test set is computed by $\frac{1}{k} \#\{y_i : f_q(x_i) \leq y_i \leq f_{1+q-\alpha}(x_i), i = 1, 2, ..k.\}$, where $\#$ denotes the cardinality of the set. Also, the MPIW can be computed by $\frac{1}{k} \sum_{i=1}^{k} |f_{1+q-\alpha}(x_i)\} - f_q(x_i)|$.

An effective PI method must achieve the target $1 - \alpha$ calibration while minimizing the PI width, as measured by the MPIW value. Furthermore, we define the Prediction Interval Coverage Error (PICE) as $\max(0, (1 - \alpha) - PICP)$ to quantify the extent to which the model falls short of the target calibration (1-$\alpha$). For comparing PI estimation models, a natural decision criterion is that the model with the lowest PICE should be considered the best. If all models successfully achieve the target calibration, the one with the minimum MPIW should be deemed the most optimal.

**Baseline methods**

To evaluate the quality of the PIs produced by the SSVQR model, it is essential to establish appropriate baseline methods for comparison. One of the key strengths of SVM machines is their ability to always obtain the global optimal solution, maintaining their relevance and applicability in modern cutting-edge technology. As detailed in Table 14, PI estimation through SVQR and LS-SVR only obtains the global optimal solution but lacks sparsity. In contrast, the SSVQR PI can obtain the optimal global solution as well as the sparse solution. In view of this, we find the SVQR and LS-SVR models are qualified enough to be compared with the SSVQR model for the PI estimation task in SVM framework. The Tube and LUBE loss PI estimation methods available in SVM framework do not guarantee the global optimal solution and their solution may vary with the choice of initialization. However, we have considered the Tube loss and an improved version of the LUBE loss, the QD loss function (Pearce et al. (2018)) in deep forecasting models to compare their performance with the SVM based probabilistic forecasting models.

**Experimental Setup and Parameter Tunning**

All SVM based PI estimation models have been implemented in MATLAB. To estimate both bounds of the PI, we utilize the RBF kernel, defined as $k(x_i, x_j) = e^{-\gamma ||x_i - x_j||^2}$. As detailed in Algorithm 1 and Algorithm 2, SVQR and SSVQR require solving the QPP (9) and LPP (14) twice to obtain the quantile bounds of the PI respectively. We solve the QPPs of the SVQR PI model and the LPPs of the SSVQR model in MATLAB using the 'quadprog' and 'linprog' packages respectively.

The 'linprog' function employs the "dual-simplex-highs" algorithm by default, which may not scale efficiently for very large datasets. A more suitable open-source alternative for solving the LPPs in the SSVQR PI model is the "HiGHS" solver, which is specifically designed to handle large-scale linear programming problems involving thousands or even millions of variables and constraints with high efficiency.

The SVQR problem(9) or SSVQR (14) problem requires the supply of the two user defined parameters $C$ and RBF kernel parameter $\gamma$ for non-linear PI estimation. We have tunned the value of the these parameters using the grid search in the search space $\{2^{-8}, 2^{-7}, ...., 2^7, 2^8\} \times \{2^{-8}, 2^{-7}, ..., 2^7, 2^8\}$.

**Artificial Datasets**

First, we generate six distinct artificial datasets. In each dataset, the values of $x_i$ are randomly sampled from a uniform distribution $U(-5, 5)$, while the corresponding $y_i$ values are obtained by polluting the function $(1 - x_i + 2x_i^2)e^{-0.5x_i^2}$ with different types of noise as described below.

$$\textbf{AD 1:-} \quad y_i = (1 - x_i + 2x_i^2)e^{-0.5x_i^2} + \xi_i, \text{where } \xi_i \sim N(0, 0.6)$$
$$\textbf{AD 2:-} \quad y_i = (1 - x_i + 2x_i^2)e^{-0.5x_i^2} + \xi_i, \quad \text{where } \xi_i \sim \chi^2(3)$$
$$\textbf{AD 3:-} \quad y_i = (1 - x_i + 2x_i^2)e^{-0.5x_i^2} + \xi_i, \text{where } \xi_i \sim N(0, 0.4)$$
$$\textbf{AD 4:-} \quad y_i = (1 - x_i + 2x_i^2)e^{-0.5x_i^2} + \xi_i, \text{where } \xi_i \sim N(0, 0.8)$$
$$\textbf{AD 5:-} \quad y_i = (1 - x_i + 2x_i^2)e^{-0.5x_i^2} + \xi_i, \text{where } \xi_i \sim U(-5, 5)$$
$$\textbf{AD 6:-} \quad y_i = (1 - x_i + 2x_i^2)e^{-0.5x_i^2} + \xi_i, \text{where } \xi_i \sim U(-4, 4)$$

In case of each dataset, 2500 data points are generated in which 1000 data points are considered for training and rest of them are considered for testing.

| | $(\bar{q},\ 1+\bar{q}-\alpha)$ | RMSE(Lw,Up) | Spar (Lw,Up) | CP (Lw,Up) | PICP | PICE | MPIW | Time (s) |
|---|---|---|---|---|---|---|---|---|
| SVQR | (0.01, 0.96) | (1.8021, 1.4446) | (0%, 0%) | (0.0110, 0.9680) | 0.957 | 0 | 2.9054 | 0.6157 |
| | (0.015, 0.965) | (1.7046, 1.4653) | (0%, 0%) | (0.0150, 0.9720) | 0.957 | 0 | 2.8244 | 0.5844 |
| | (0.02, 0.97) | (1.6519, 1.4799) | (0%, 0%) | (0.0180, 0.9720) | 0.954 | 0 | 2.7855 | 0.6169 |
| | (0.025, 0.975) | (1.5857, 1.5016) | (0%, 0%) | (0.0220, 0.9760) | **0.954** | 0 | **2.7379** | 0.5546 |
| | (0.03, 0.98) | **(1.5030, 1.5667)** | (0%, 0%) | (0.0250, 0.9800) | 0.955 | 0 | 2.7212 | **0.5150** |
| SSVQR | (0.01, 0.96) | (1.8060, 1.4418) | (15%, 18%) | (0.0110, 0.9680) | 0.957 | 0 | 2.9056 | 0.4783 |
| | (0.015, 0.965) | (1.7051, 1.4714) | (15%, 18%) | (0.0150, 0.9700) | 0.955 | 0 | 2.8275 | 0.5485 |
| | (0.02, 0.97) | (1.6497, 1.4861) | (15%, 18%) | 0.0160, 0.9720 | 0.956 | 0 | 2.7939 | **0.4712** |
| | (0.025, 0.975) | **(1.5517, 1.4978)** | (15%, 20%) | (0.0250, 0.9750) | **0.950** | 0 | **2.6995** | 0.4928 |
| | (0.03, 0.98) | (1.5133, 1.5604) | **(16%, 20%)** | (0.0290, 0.9800) | 0.951 | 0 | 2.7288 | 0.6054 |

Table 15: Performance of the SVQR and SSVQR PI models on AD 1 dataset. Lw: Lower, Up: Upper, Spar: Sparsity, Time(s) : Training time in seconds. SSVQR PI models always obtain the sparse solution. Also, for $\bar{q} = 0.025$, SSVQR model obtains the best quality PI with MPIW = 2.70 and PICP = 0.95.

| | $(\bar{q},\ 1+\bar{q}-\alpha)$ | RMSE(Lw,Up) | Spar (Lw,Up) | CP (Lw,Up) | PICP | PICE | MPIW | Time (s) |
|---|---|---|---|---|---|---|---|---|
| SVQR | (0.01, 0.96) | **(4.2653, 5.7073)** | (0%, 0%) | (0.0100, 0.9470) | 0.937 | 0.013 | 8.3755 | 0.805 |
| | (0.015, 0.965) | (4.1993, 6.0698) | (0%, 0%) | (0.0210, 0.9520) | 0.931 | 0.019 | 8.6865 | 0.699 |
| | (0.02, 0.97) | (4.1312, 6.5751) | (0%, 0%) | (0.0220, 0.9610) | **0.939** | **0.011** | **9.1619** | 0.6595 |
| | (0.025, 0.975) | (4.0525, 6.8163) | (0%, 0%) | (0.0340, 0.9650) | 0.931 | 0.019 | 9.3256 | 0.719 |
| | (0.03, 0.98) | (4.0251, 7.4745) | (0%, 0%) | (0.0350, 0.9720) | 0.937 | 0.013 | 9.9780 | 0.6809 |
| SSVQR | (0.01, 0.96) | **(4.1628, 5.5481)** | (20%, 15%) | (0.0080, 0.9390) | 0.931 | 0.019 | 8.1036 | **0.502** |
| | (0.015, 0.965) | (4.0206, 5.9335) | (18%, 18%) | (0.0100, 0.9520) | 0.942 | 0.008 | 8.3460 | 0.5173 |
| | (0.02, 0.97) | (4.0156, 6.3077) | **(20%, 25%)** | (0.0110, 0.9640) | **0.953** | 0 | **8.7801** | 0.5128 |
| | (0.025, 0.975) | (3.9444, 7.1596) | (15%, 20%) | (0.0240, 0.9680) | 0.944 | 0.006 | 9.5723 | 0.4694 |
| | (0.03, 0.98) | (3.9309, 7.3834) | (20%, 20%) | (0.0290, 0.9740) | 0.945 | 0.005 | 9.8075 | 0.5113 |

Table 16: Performance of the SVQR and SSVQR PI models on AD 2 dataset. Lw: Lower, Up: Upper, Spar: Sparsity, Time(s) : Training time in seconds. SSVQR PI models always obtain the sparse solution. Also, for $\bar{q} = 0.02$, SSVQR model obtains the best quality PI with PCIP = 0.953 and MPIW = 8.78.

| | $(\bar{q},\ 1+\bar{q}-\alpha)$ | RMSE(Lw,Up) | Spar (Lw,Up) | CP (Lw,Up) | PICP | PICE | MPIW | Time (s) |
|---|---|---|---|---|---|---|---|---|
| SVQR | (0.01, 0.96) | (1.7894, 1.5837) | (0%, 0%) | ( 0.0120, 0.9570) | 0.945 | 0.050 | 2.9000 | 0.4872 |
| | (0.015, 0.965) | (1.7355, 1.6297) | (0%, 0%) | (0.0130, 0.9590) | **0.946** | 0.004 | **2.8870** | 0.5077 |
| | (0.02, 0.97) | **(1.5696, 1.6407)** | (0%, 0%) | (0.0230, 0.9590) | 0.936 | 0.014 | 2.7223 | 0.5004 |
| | (0.025, 0.975) | (1.5070, 1.6927) | (0%, 0%) | (0.0280, 0.9680) | 0.940 | 0.010 | 2.7100 | 0.4907 |
| | (0.03, 0.98) | (1.4694, 1.7477) | (0%, 0%) | (0.0300, 0.9750) | 0.945 | 0.005 | 2.7394 | **0.4745** |
| SSVQR | (0.01, 0.96) | 1.8411, 1.5781 | (20%, 40%) | 0.0120, 0.9570 | 0.945 | 0.005 | 2.9462 | **0.4367** |
| | (0.015, 0.965) | (1.8297, 1.5917) | (40%, 30%) | (0.0120, 0.9570) | **0.945** | 0.005 | **2.9458** | 0.4563 |
| | (0.02, 0.97) | (1.6999, 1.6969) | (40%, 40%) | (0.0180, 0.9600) | 0.942 | 0.008 | 2.9026 | 0.4671 |
| | (0.025, 0.975) | (1.6509, 1.7191) | **(40%, 40%)** | (0.0220, 0.9620) | 0.940 | 0.010 | 2.8817 | 0.4501 |
| | (0.03, 0.98) | **(1.4865, 1.7721)** | (30% , 40%) | (0.0290, 0.9700) | 0.941 | 0.009 | 2.7739 | 0.4717 |

Table 17: Performance of the SVQR and SSVQR PI models on AD 3 dataset. Lw: Lower, Up: Upper, Spar: Sparsity, Time(s) : Training time in seconds. SSVQR PI models always obtain the sparse solution. SSVQR and SVQR model tends to obtain the similar performance.

| | $(\bar{q},\ 1+\bar{q}-\alpha)$ | RMSE(Lw,Up) | Spar (Lw,Up) | CP (Lw,Up) | PICP | PICE | MPIW | Time (s) |
|---|---|---|---|---|---|---|---|---|
| SVQR | (0.01, 0.96) | (2.5953, 2.1197) | (0%, 0% ) | (0.0160, 0.9530) | 0.937 | 0.013 | 4.1549 | 0.723 |
| | (0.015, 0.965) | (2.5595, 2.1292) | (0%, 0%) | (0.0160, 0.9530) | 0.937 | 0.013 | 4.1268 | 0.6713 |
| | (0.02, 0.97) | (2.4221, 2.2193) | (0%, 0%) | (0.0200, 0.9610) | **0.941** | 0.009 | **4.0817** | 0.6704 |
| | (0.025, 0.975) | **(2.2561, 2.2555)** | (0%, 0%) | (0.0290, 0.9630) | 0.934 | 0.016 | 3.9365 | 0.6815 |
| | (0.03, 0.98) | (2.2211, 2.3699) | (0%, 0%) | (0.0310, 0.9670) | 0.936 | 0.014 | 4.0273 | **0.6343** |
| SSVQR | (0.01, 0.96) | (2.5842, 2.1364) | (20%, 40%) | (0.0160, 0.9530) | 0.937 | 0.013 | 4.1577 | 0.4704 |
| | (0.015, 0.965) | (2.5475, 2.1911) | (20%, 20%) | (0.0160, 0.9560) | 0.940 | 0.010 | 4.1802 | 0.4745 |
| | (0.02, 0.97) | (2.4939, 2.2455) | (20%, 20%) | (0.0190, 0.9610) | **0.942** | 0.008 | **4.1830** | 0.4818 |
| | (0.025, 0.975) | **(2.3617, 2.3037)** | **(40%, 20%)** | (0.0240, 0.9640) | 0.940 | 0.010 | 4.1002 | **0.4662** |
| | (0.03, 0.98) | (2.2689, 2.4147) | (40%, 20%) | (0.0290, 0.9700) | 0.941 | 0.009 | 4.1210 | 0.4884 |

Table 18: Performance of the SVQR and SSVQR PI models on AD 4 dataset. Lw: Lower, Up: Upper, Spar: Sparsity, Time(s) : Training time in seconds. SSVQR PI models always obtain the sparse solution. SSVQR and SVQR model tends to obtain the similar performance.

| | $(\bar{q},\ 1+\bar{q}-\alpha)$ | RMSE(Lw,Up) | Spar (Lw,Up) | CP (Lw,Up) | PICP | PICE | MPIW | Time (s) |
|---|---|---|---|---|---|---|---|---|
| | (0.01, 0.96) | (4.9694, 4.4016) | (0%, 0% ) | **(0.0100, 0.9560)** | **0.946** | 0.004 | **8.1044** | 0.7917 |
| | (0.015, 0.965) | (4.8456, 4.4539) | (0%, 0%) | (0.0180, 0.9600) | 0.942 | 0.008 | 8.0257 | 0.8009 |
| SVQR | (0.02, 0.97) | (4.8100, 4.4860) | (0%, 0%) | (0.0190, 0.9620) | 0.943 | 0.007 | 8.0223 | 0.736 |
| | (0.025, 0.975) | **(4.7111, 4.5143)** | (0%, 0%) | (0.0280, 0.9640) | 0.936 | 0.014 | 7.9430 | 0.6895 |
| | (0.03, 0.98) | (4.6921, 4.5368) | (0%, 0%) | (0.0290, 0.9680) | 0.939 | 0.011 | 7.9481 | **0.6191** |
| | (0.01, 0.96) | (5.2658, 4.5044) | (40%, 30%) | **(0.0050, 0.9610)** | **0.956** | 0.00 | **8.5344** | 0.5394 |
| | (0.015, 0.965) | (4.8787, 4.5073) | (35%, 30%) | (0.0160, 0.9610) | 0.945 | 0.005 | 8.1171 | 0.4835 |
| SSVQR | (0.02, 0.97) | **(4.8138, 4.5249)** | (30%, 30%) | (0.0190, 0.9650) | 0.946 | 0.004 | 8.0659 | **0.472** |
| | (0.025, 0.975) | (4.7464, 4.6037) | (25%, 30%) | (0.0260, 0.9670) | 0.941 | 0.009 | 8.0692 | 0.5102 |
| | (0.03, 0.98) | (4.6962, 4.7637) | **(30%, 40%)** | (0.0290, 0.9720) | 0.943 | 0.007 | 8.1832 | 0.5068 |

Table 19: Performance of the SVQR and SSVQR PI models on AD 5 dataset. Lw: Lower, Up: Upper, Spar: Sparsity, Time(s) : Training time in seconds. SSVQR PI models always obtain the sparse solution. For $\bar{q} = 0.01$, SSVQR PI model obtains the best quality PI with PICP =0.956 and MPIW = 8.53.

| | $(\bar{q},\ 1+\bar{q}-\alpha)$ | RMSE(Lw,Up) | Spar (Lw,Up) | CP (Lw,Up) | PICP | PICE | MPIW | Time (s) |
|---|---|---|---|---|---|---|---|---|
| | **(0.01, 0.96)** | (6.1510, 5.5057) | (0%, 0%) | **(0.0110, 0.9630)** | **0.952** | 0 | **10.0826** | **0.6437** |
| | (0.015, 0.965) | (6.0517, 5.5584) | (0%, 0%) | (0.0150, 0.9670) | 0.952 | 0 | 10.031 | 0.6798 |
| SVQR | (0.02, 0.97) | (5.9777, 5.6130) | (0%, 0% ) | (0.0170, 0.9700) | 0.953 | 0 | 10.0141 | 1.0752 |
| | (0.025, 0.975) | (5.8879, 5.6868) | (0%, 0%) | (0.0230, 0.9760) | 0.953 | 0 | 9.9987 | 0.9265 |
| | (0.03, 0.98) | **(5.7599, 5.7156)** | (0%, 0%) | (0.0290, 0.9780) | 0.949 | 0.001 | 9.8878 | 0.9445 |
| | (0.01, 0.96) | (5.9304, 5.4581) | (10%, 15%) | (0.0110, 0.9610) | 0.95 | 0 | 9.7944 | 0.569 |
| | (0.015, 0.965) | (5.9107, 5.5686) | (10%, 20%) | (0.0140, 0.9700) | 0.956 | 0 | 9.9016 | 0.5656 |
| SSVQR | **(0.02, 0.97)** | (5.7836, 5.5880) | (15%, 25%) | **(0.0180, 0.9700)** | **0.952** | 0 | **9.7807** | 0.5766 |
| | (0.025, 0.975) | **(5.6486, 5.6038)** | **(20%, 20%)** | (0.0260, 0.9710) | 0.945 | 0.005 | 9.6443 | **0.5423** |
| | (0.03, 0.98) | (5.6184, 5.6506) | (10%, 15%) | (0.0300, 0.9760) | 0.946 | 0.004 | 9.6653 | 0.5636 |

Table 20: Performance of the SVQR and SSVQR PI models on AD 6 dataset. Lw: Lower, Up: Upper, Spar: Sparsity, Time(s):Training time in seconds. SSVQR PI models always obtain the sparse solution. For $\bar{q} = 0.01$, SSVQR PI model obtains the best quality PI with PICP =0.956 and MPIW = 8.53.

### Artificial Datasets Results, Discussion and Analysis

We present the performance of the SVQR and SSVQR model for PI estimation task with the different values of $\bar{q}$ for each of simulated dataset in Table 15-20. The leftmost column of these Tables list the different pairs of target quantiles $(\bar{q},\ 0.95 + \bar{q})$, required for the PI estimation. As detailed in Section 5.1 of this paper, for artificial datasets, the quality of the estimated upper and lower quantiles of the PI can be best evaluated by computing the RMSE between the estimated quantiles and their corresponding true quantile functions. The third column of the Tables 15-20 list these RMSE for different values of $\bar{q}$.

To effectively illustrate the comparative performance between the SVQR and SSVQR models across various artificial datasets, we normalize the MPIW values by dividing them by the variance of each dataset (AD1–AD6). Additionally, we compute the aggregated RMSE as the mean of RMSE (Lp) and RMSE (Up), which represent the RMSE values corresponding to the lower and upper quantile bounds of the PI, respectively. Further, we also normalize them by dividing the variance present in each datasets for the comparative study.

Figure 10 shows the comparison of the quality of PIs estimated by the SVQR and SSVQR models. For a high-quality PI, the PICP should be greater than or equal to the target value of $(1 - \alpha) = 0.95$, while maintaining the smallest possible MPIW values. Among the six datasets (AD1-AD6), the SSVQR-based PI model successfully achieves the target PICP of 0.95 on four datasets, where as the SVQR-based PI model fails to meet the target PICP on four datasets. Figure 11 illustrates the comparisons of the normalized aggregated RMSE obtained by the SVQR and SSVQR models. The quantile bounds estimated by the SSVQR model are comparable to, or slightly better than, those obtained from the SVQR model.

Replacing $L_2$ regularization with $L_1$ regularization in the SVM quantile regression model results in improved PI quality. However, the major advantage of using the SSVQR model over SVQR model is its ability to obtain the sparse solution vector. Figure 12(a) plots the sparsity of the solution vector corresponding to the upper and lower quantile bounds obtained by the SSVQR model. It highlights that the SSVQR model effectively

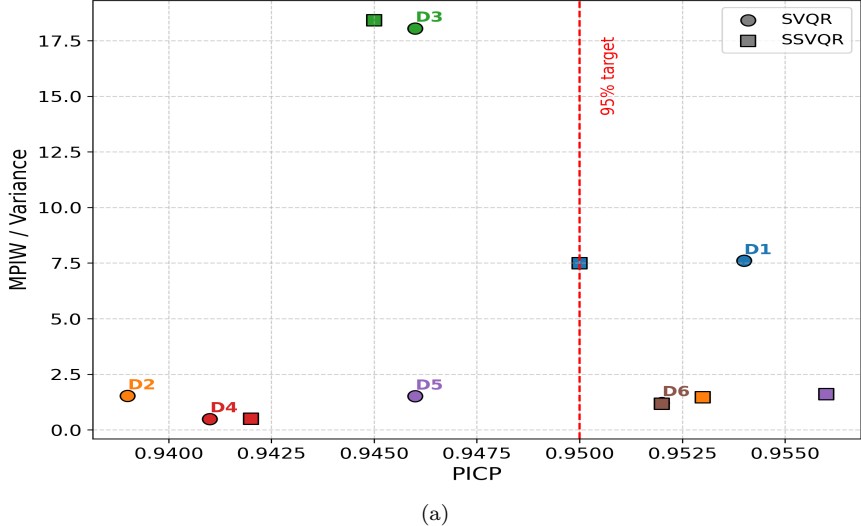

(a)

Figure 10: Comparison of the quality of PIs estimated by the SVQR and SSVQR models. For a high-quality PI, the PICP should be greater than or equal to the target value of $(1 - \alpha) = 0.95$, while maintaining the smallest possible MPIW values. Among the six datasets (AD1–AD6), the SSVQR-based PI model successfully achieves the target PICP of 0.95 on four datasets, whereas the SVQR-based PI model fails to meet the target PICP on four datasets.

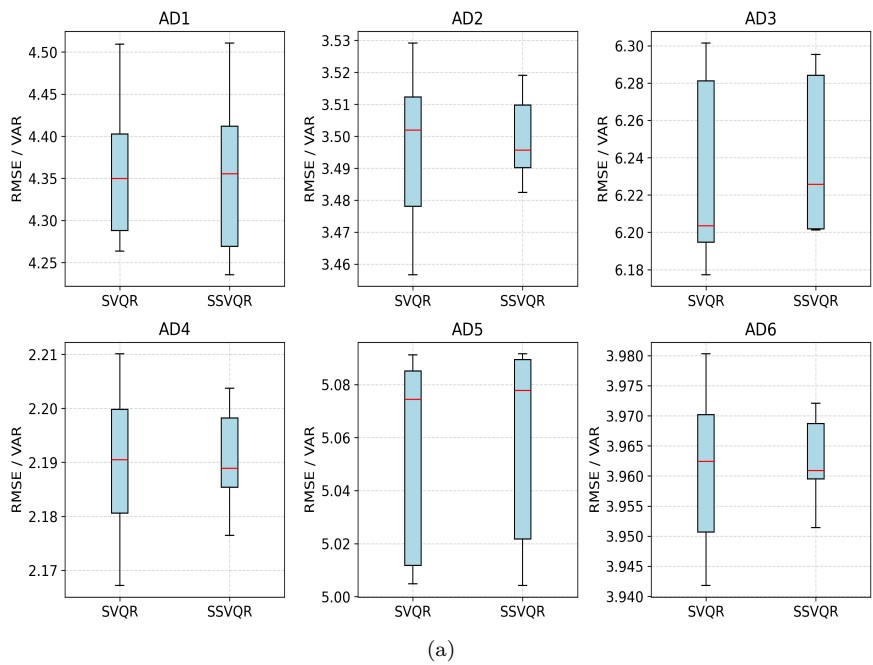

(a)

Figure 11: Comparison of the quality of the quantile functions estimated by the SVQR and SSVQR models is performed using the aggregated RMSE metric. The aggregated RMSE is computed as the mean of RMSE(Lw) and RMSE(up) which represent the RMSE values for the lower and upper quantile bounds of the PI, respectively. These RMSE values are then normalized by dividing by the variance of each dataset.

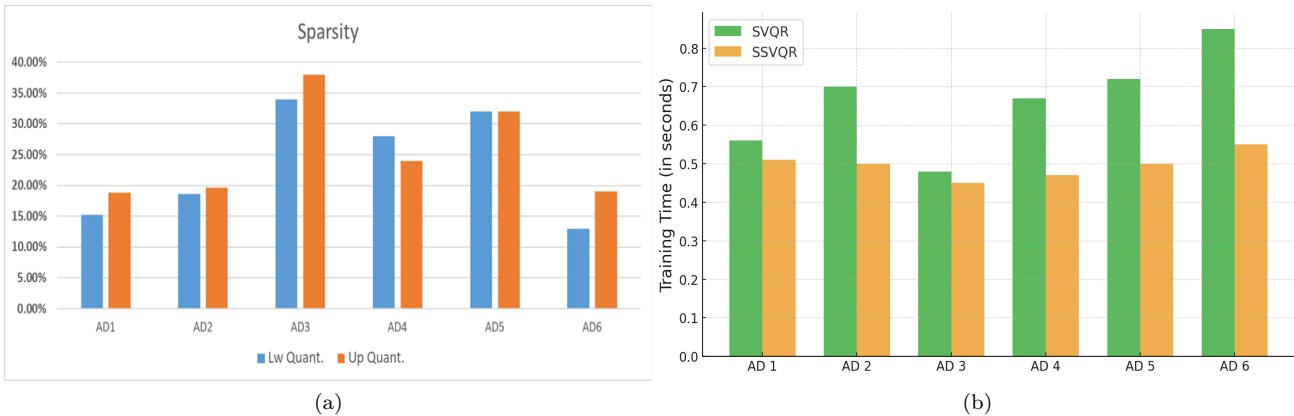

(a)                                    (b)

Figure 12: (a) Qunatile functions estimated by the SSVQR model is sparse while SVQR model fails to obtain the sparse solution. (b) Average training time comparison of the SVQR and SSVQR models for PI estimation.

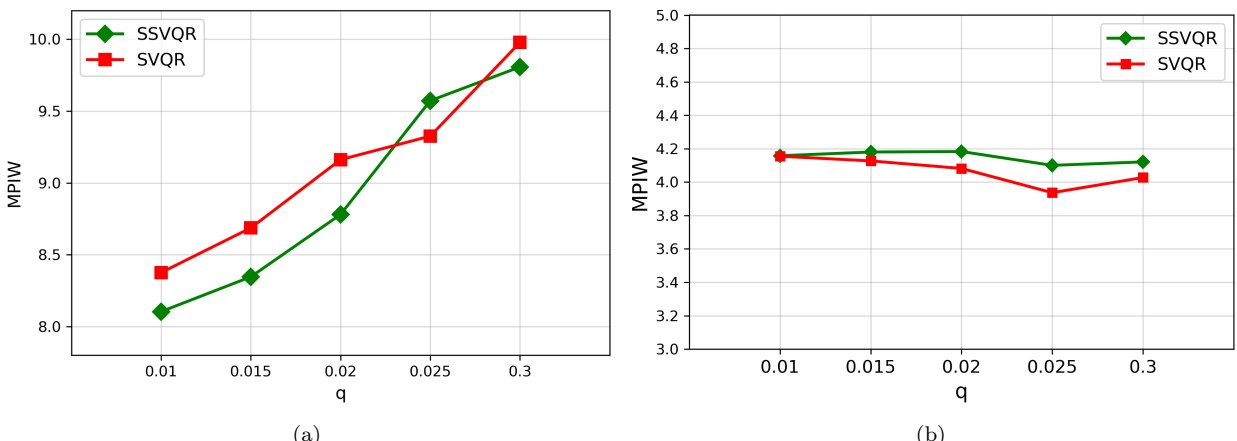

(a)                                    (b)

Figure 13: Plot of the MPIW values obtained by the SVQR and SSVQR PI models as a function of $\bar{q}$ for (a) the AD2 dataset and (b) the AD4 dataset. The AD2 dataset contains asymmetric noise drawn from a $\chi^2$ distribution, resulting in a higher concentration of data points in the lower region of the input–target space. Consequently, the MPIW increases as the PI tube shifts upward with larger values of $\bar{q}$. In contrast, the AD4 dataset contains noise from a normal distribution; therefore, the centered PI obtained with the $\bar{q} = 0.025$ yields the narrower PI.

reduces significant coefficients of the solution vector to near zero which enables the feature selection task in PI estimation and also simplify the overall prediction process.

We compare the overall average training time (in seconds) taken by the SVQR PI and SSVQR PI models for the PI estimation task across different artificial datasets in Figure 12(b). It shows that the SSVQR requires fewer seconds train the PI model than the SVQR model. We have solved the QPPs of the SVQR PI model and LPPs of the SSVQR model in MATLAB with 'quadprog' and 'linprog' packages respectively.

Another key observation from the numerical results in Tables 15-20 is the consistent performance of the SVQR and SSVQR PI models across different datasets. In all cases, both the SVQR and SSVQR PI models manage to approximately achieve 95% target coverage. We plot the MPIW values obtained by the SSVQR and SVQR PI models against different values of $\bar{q}$ on dataset AD2 and AD4 in Figure 13. On AD2 dataset, the MPIW values of the PI obtained by both models increase with $\bar{q}$. It is evident from Table 16 that this

increase is not related to the PICP values obtained by the SVQR and SSVQR PI models. Actually, it is caused by the nature of noise present in the AD2 dataset. The AD2 dataset contains asymmetric noise from the ($\chi^2$) distribution, leading to a higher density of data points in the lower region of the input-target space, which gradually decreases as we move upward. As $\bar{q}$ decreases, the resultant PI shifts downward, passing through denser regions of the data cloud, leading to lower MPIW values. It shows that the movement of the PI tube due to change of $\bar{q}$ values may lead to the narrower PI particularly in presence of the asymmetric noise in the data. Apart from the AD2 dataset, all other artificial datasets contain noise from symmetric distributions. In these datasets, the centered PI (($f_{0.025}(x), f_{0.975}(x)$) is expected to achieve the high quality PI. Figure 13(b) shows that at $\bar{q} = 0.025$, both the SVQR and SSVQR PI models attain the lowest MPIW values on AD4 dataset.

### SVM PI estimation methods on benchmark datasets

We have done experiments on the two popular benchmark datasets namely Boston Housing (Harrison Jr & Rubinfeld (1978)) and Concrete (Yeh (1998)) and evaluate the quality of the PI estimated by the SSVQR, SVQR and LS-SVR based PI estimation methods for different value of the $\bar{q}$ with non-linear RBF kernel. Table 21 and 22 contains the numerical results obtained on the Boston Housing and Concrete datasets respectively. We can observe that the SSVQR, SVQR and LS-SVR PI models obtains a similar quality of the PI but, SSVQR models always obtain the sparse solution vector.

| | q | Spar (Lw, Up) | CP (Lw, Up) | PICP | PICE | MPIW | Time (s) |
|---|---|---|---|---|---|---|---|
| | (0.025, 0.925) | (0%, 0%) | (0.028, 0.930) | **0.90** | 0 | **28.55** | **0.1347** |
| SVQR | (0.05, 0.95) | (0%, 0%) | (0.052, 0.950) | 0.90 | 0.00 | 33.06 | 0.1472 |
| | (0.075, 0.975) | (0%, 0%) | (0.088, 0.972) | 0.88 | 0.02 | 38.23 | 0.1670 |
| | **(0.025, 0.925)** | **(17%, 16%)** | (0.027, 0.927) | **0.90** | 0 | **28.62** | **0.0657** |
| SSVQR | (0.05, 0.95) | (15%, 22%) | (0.048, 0.947) | 0.90 | 0 | 32.77 | 0.0678 |
| | (0.075, 0.975) | (14%, 28%) | (0.080, 0.967) | 0.89 | 0.01 | 38.24 | 0.0681 |
| | (0.025, 0.925) | (0%,0% ) | | 0.91 | 0 | 31.19 | **0.0064** |
| LS-SVR | **(0.050, 0.950)** | (0%,0%) | | **0.91** | 0 | **30.18** | 0.0067 |
| | (0.075, 0.975) | (0%,0%) | | 0.89 | 0.01 | 31.19 | 0.0060 |

Table 21: Comparison of different SVM PI estimation methods on Boston Housing. Lw: Lower, Up: Upper, Spar: Sparsity, Time(s) : Training time in seconds.

| | q | Spar (Lw, Up) | CP (Lw, Up) | PICP | PICE | MPIW | Time (s) |
|---|---|---|---|---|---|---|---|
| | (0.025, 0.925) | (0%, 0%) | (0.0219, 0.8997) | 0.8777 | 0.0723 | 32.5492 | 2.6141 |
| SVQR | **(0.05, 0.95)** | (0%, 0%) | (0.0408, 0.9436) | **0.9028** | **0.0472** | **28.6541** | 2.8405 |
| | (0.075, 0.975) | (0%, 0%) | (0.0690, 0.9561) | 0.8871 | 0.0629 | 32.1635 | **2.5197** |
| | (0.025, 0.925) | (12%, 12%) | (0.0340, 0.9029) | 0.8689 | 0.0811 | 29.4962 | 0.1858 |
| SSVQR | (0.05, 0.95) | (12%, 12%) | (0.0583, 0.9272) | 0.8689 | 0.0811 | **27.8917** | **0.1795** |
| | **(0.075, 0.975)** | **12%, 12%** | (0.0777, 0.9515) | **0.8738** | 0.0762 | 30.2547 | 0.1837 |
| | (0.025, 0.925) | 0%,0% | | 0.9126 | 0.0374 | 28.2429 | **0.4125** |
| LS-SVR | (0.050, 0.950) | (0%,0%) | | 0.8932 | 0.0568 | **27.3308** | 0.4631 |
| | **(0.075, 0.975)** | (0%,0%) | | **0.9029** | 0.0471 | 28.2429 | 0.3954 |

Table 22: Comparison of different SVM PI estimation methods on Concrete dataset. Lw: Lower, Up: Upper, Spar: Sparsity, Time(s) : Training time in seconds.

