# OpenReview forum: "Support Vector Machines : A more certain estimate of uncertainty"
_TMLR — Under review for TMLR_

### Review · Reviewer_V2Yf · 2026-04-28

**Summary Of Contributions:**

# Contributions

The paper revisits support vector machines as a backbone for uncertainty quantification in regression, arguing that SVM-based methods can provide more stable prediction intervals and probabilistic forecasts than neural-network or random-forest alternatives, especially in small- and moderate-data regimes. It surveys existing SVM-based UQ approaches, formulates prediction-interval estimation using support vector quantile regression, proposes a sparse SVQR variant intended to support feature selection, adapts SVQR to a split-conformal regression setting, and empirically compares these methods against NN, RF, and deep probabilistic forecasting baselines across toy, tabular, conformal, and time-series experiments.

# Strengths

The paper raises a practically relevant point: UQ methods should be evaluated not only by coverage and interval width, but also by the stability of the estimated intervals across repeated training runs. Its emphasis on small-data settings is well motivated, since overparameterized neural models can be unstable or unnecessarily complex in such regimes. The paper also usefully connects several threads (quantile regression, SVMs, sparsity, feature selection, conformal prediction, and probabilistic forecasting) under a unified SVM-based perspective. The empirical results are directionally interesting: they suggest that deterministic SVQR models can produce reproducible intervals and conformal sets, sometimes with competitive PICP/MPIW, while sparse variants may reduce feature dimensionality. Overall, the work has promising framing and could make a useful case for SVMs as strong small-data UQ baselines, **even though the current version needs stronger theory and experiments to support its broader claims**.


# Weaknesses

I have several concerns about the proposed theory and experimental results in this paper. I am listing them in the following section.

**Audience:**

Yes

**Audience Explanation:**

Uncertainty quantification is a very important problem in machine learning. The paper's claims about model uncertainty, and the proposed fixes with SVMs are interesting. This can also be important to a broader community, who may think about how to make uncertainty quantification itself more deterministic in a wide range of settings.

**Claims And Evidence:**

No

**Claims Explanation:**

# Concerns about theory

## The invalid sparse-SVQR rewrite


The paper defines the sparse SVQR primal problem as


$$
\min_{w,b}\; \frac{1}{2}\lVert w \rVert_1 + C\sum_{i=1}^m \rho_q\bigl(y_i-(w^\top \phi(x_i)+b)\bigr). \tag{7}
$$


It then claims that, using the kernel trick / representer theorem, this can be rewritten as an optimization in kernel coefficients $u$ with an $\ell_1$ penalty on $u$.


If


$$
f(x)=w^\top \phi(x)+b = \sum_{j=1}^m u_j k(x_j,x)+b,
$$


then necessarily


$$
w = \sum_{j=1}^m u_j \phi(x_j).
$$


Therefore the original regularizer becomes


$$
\lVert w \rVert_1 = \left\lVert \sum_{j=1}^m u_j \phi(x_j) \right\rVert_1,
$$


**not** $\lVert u \rVert_1$.


So the problem in $(7)$ is **not** equivalent to a problem of the form


$$
\min_{u,b}\; \frac{1}{2}\lVert u \rVert_1 + C\sum_{i=1}^m \rho_q\Bigl(y_i-\bigl(\sum_{j=1}^m k(x_j,x_i)u_j+b\bigr)\Bigr).
$$

Equation (9) optimizes an $\ell_1$ penalty on the **kernel expansion coefficients** $u$, not on the **feature-space weight vector** $w$. These are different notions of sparsity:


- sparsity in $w$ = feature sparsity;
- sparsity in $u$ = support-vector / expansion-coefficient sparsity.


Hence the paper's derivation does **not** justify the claimed feature-selection interpretation. Moreover, the usual representer theorem does not automatically justify this step: it applies cleanly to regularizers that are nondecreasing functions of the RKHS norm (e.g. $\lVert w \rVert_{\mathcal H}^2$), not to a coordinatewise $\ell_1$ penalty on $w$.


## Quantile-index range error


The paper's introduction states the correct condition: for a target interval of level $1-\alpha$, one may use


$$
\bigl(f_q(x), f_{1+q-\alpha}(x)\bigr)
\quad\text{for some}\quad
0\le q\le \alpha.
$$


However, the later algorithms (Algorithm 2 and 3) use the incorrect range


$$
\bar q\in(0,1-\alpha).
$$


This appears in:


- page 8, Algorithm 2: feature selection through SSVQR;
- page 8, Algorithm 3: conformal regression through SVQR.


The algorithms are mathematically incorrect as written. They should use


$$
\bar q\in[0,\alpha]
$$


instead of


$$
\bar q\in(0,1-\alpha).
$$


The experiments may still be valid if they used standard central choices such as $\bar q=\alpha/2$, but the stated algorithms contain a quantile-indexing error.

# Concerns about experiments

I would assess the experiments as interesting but not sufficient for the claims. The experiments support a narrow point, that SVM-based UQ estimates are deterministic under fixed data/hyperparameters, but they do not convincingly show that SVMs give better uncertainty quantification overall.

**What the experiments do show**

SVM/SVQR methods have lower run-to-run variability than NN/RF methods when the dataset, split, hyperparameters, and solver setup are fixed. This is visible in the toy PI experiment, the conformal experiment, and the forecasting table where SVM-based methods report $\sigma^2 = 0$.

**Where the experiments could improve**

Zero run-to-run variance is almost guaranteed by the setup. If SVQR is implemented deterministically and the data, hyperparameters, split, kernel, and solver settings are fixed, then the same output across runs is expected. The experiment confirms this, but it is not a deep empirical discovery. The important question is not “does the SVM change across reruns?” The important question is whether the SVM intervals are better calibrated, sharper, more conditionally valid, more robust to split variation, and more useful under distribution shift. The experiments mostly do not test those harder questions. The paper equates stability with reliability. A deterministic interval can be reproducible and still wrong.

**Specific concerns**

1. The paper defines model uncertainty as a sum of squared deviations over test points and runs. That depends on the number of test points, the scale of the target, and the number of repetitions. I think some sort of normalization would make the numbers comparable across different settings.

2. **The metric favors deterministic methods by construction**. Any deterministic method gets $\sigma^2 = 0$. That does not mean it has no epistemic uncertainty; it only means the training algorithm returns the same function under the same setup.

3. **This is the most important of my concerns.** The paper says all models are trained across 10 runs using identical training and calibration splits. That isolates training randomness, but ignores split randomness. They should run many random train/calibration/test splits and report distributions of PICP, MPIW, interval score, and conditional coverage. The current experiment only shows that deterministic SVQR is deterministic after the split is fixed.

4. Should include ablations over the neural network architecture choice/provide better justification.

5. No Gaussian process or Bayesian baselines. Since the paper is about small-data UQ, Gaussian processes are an obvious baseline. Their absence weakens the experimental case.

6. The forecasting experiments are not convincing to me. The datasets are too small. While I understand the point that the authors are trying to make is exactly about small dataset cases, the authors should also run their method against the baselines in large dataset scenarios, and report how they fare against standard baselines. Otherwise, it is basically saying Neural networks are worse when data is scarce, which makes sense but is less interesting. (**It is completely fair if their method underperforms neural networks in more complex/data-rich scenarios, but a comparison should be made in the paper for scientific rigor**)

7. What is the relationship with Lasso Quantile Regression?

8. What happens for very large datasets, does kernel methods remain scalable?

9. What happens under distribution shift? I.e., how does this method perform against the baselines if we have covariate shift or other types of shifts?

**Requested Changes:**

Aside from addressing my concerns mentioned in the weaknesses, I would like to mention some minor issues:

## Writing issues that can be improved

1. Citations should use ~\citep{reference} or ~\citet{reference} appropriately: in-line citations in this paper seems to have been formatted incorrectly.

2. Writing is poor: the introduction feels more like related works. Prior work should be summarized more carefully and distinctions with this paper’s approach should be emphasized more. Bulk of the prior works in the introduction should be moved towards the related works section.

3. Figure 1 is not even contained within the page layout.

---

> ### Author Response · Authors · 2026-07-13
>
> We gratefully acknowledge the reviewers’ thoughtful and constructive comments and believe that careful consideration of these suggestions has meaningfully strengthened the quality and clarity of our work. We have attached a response letter as supplementary material in our submission that presents our point-by-point responses to the reviewers’ observations and recommendations. A copy of the revised manuscript is attached with this response letter, where all changes have been highlighted in blue for ease of reference. We have significantly revised our manuscript in the light of reviewer's valuable comments and incorporated reviewer suggestions in the revised manuscript. Please see the response letter in supplementary material. We shall be happy to answer any future concerns or queries.

---

### Review · Reviewer_hCZa · 2026-05-04

**Summary Of Contributions:**

The paper deals with uncertainty quantification using support vector machines. The authors describe their own contribution as
`Our major contributions include the development of sparse SVM models for PI estimation and probabilistic forecasting, an investigation of the role of feature selection in PI estimation, and the extension of SVM regression to the Conformal Regression (CR) setting to construct more stable prediction sets with finite-sample guarantees.`

**Additional Comments:**

* In the discussion of existing NN methods for quantifying uncertainty—and particularly when distinguishing between aleatory and epistemic uncertainty—I believe [4,5,6] should also be mentioned

* "sin" should not be italized,

* There are many wrong or unusual lower case spellngs in the bibliography,

[4] Depeweg et al., Decomposition of uncertainty in Bayesian deep learning for efficient and risk-sensitive learning, International Conference on Machine Learning, 2018\
[5] Abdar et al., A review of uncertainty quantification in deep learning: Techniques, applications and challenges, Information Fusion, 2021\
[6] Hüllermeier and Waegeman ,Aleatoric and epistemic uncertainty in machine learning: An introduction to concepts and methods, Machine learning, 2021

**Audience:**

Yes

**Audience Explanation:**

Perhaps some of these considerations and findings will be of interest to some readers.

After all, this is a timely topic that is far from settled.

**Broader Impact Concerns:**

None.

**Claims And Evidence:**

No

**Claims Explanation:**

It is claimed that `SVM-based methods are more stable, sparse, and interpretable`, but there is no convincing demonstration that they are truly “more interpretable" or how interpretability is measured.

A 20% validation set and no test set for reliably assessing generalization ability seems quite insufficient to me.

The claim that `Extensive numerical experiments highlight that SVM-based UQ methods yield PIs and probabilistic forecasts that are less uncertain than those produced by modern complex deep learning and neural network models, particularly for small and moderate-scale datasets` is, in my opinion, far too general. Only a limited set of tasks was considered, and not all conceivable “complex deep learning and neural network models” were examined.

**Requested Changes:**

* Some claims need to be phrased more carefully, more specifically, and less generally.

* I recommend using a test set that is as large as possible, which is not used during the determination of all parameters and hyperparameters.

* The claim of interpretability should either be dropped or justified in detail, including an explanation of what is meant by interpretability. See, e.g., [1].

* The method is not reproducible based on the text. This problem must be resolved.

* It is claimed that “However, there is only a limited literature addressing UQ in SVM-based prediction,” yet existing works such as [2,3] are not mentioned.

* The language needs improvement. Articles are often missing, e.g.,
“In distribution-free setting” -> “In a distribution-free setting”

* The term “Conformal Regression” should be explained or supported by a reference.


[1] Doshi-Velez and Kim, Towards a rigorous science of interpretable machine learning, arXiv preprint, 2017\
[2] Wang and Pardalos, A Survey of Support Vector Machines with Uncertainties, Annals of Data Science, 2015\
[3] Xiong et al., Support Vector Machines With Uncertainty Option and Incremental Sampling for Kriging, Expert Systems,  2025

---

> ### Author Response · Authors · 2026-07-13
>
> We gratefully acknowledge the reviewers’ thoughtful and constructive comments and believe that careful consideration of these suggestions has meaningfully strengthened the quality and clarity of our work. We have attached a response letter as supplementary material in our submission that presents our point-by-point responses to the reviewers’ observations and recommendations. A copy of the revised manuscript is attached with this response letter, where all changes have been highlighted in blue for ease of reference. We have significantly revised our manuscript in the light of reviewer's valuable comments and incorporated reviewer suggestions in the revised manuscript. Please see the response letter in supplementary material. We shall be happy to answer any future concerns or queries.

---

### Review · Reviewer_MoP7 · 2026-06-17

**Summary Of Contributions:**

The paper argues that SVMs are underappreciated for uncertainty quantification in regression, mainly because they produce deterministic solutions (no run-to-run variance), can yield sparse quantile estimates via L1 regularization, and extend naturally to conformal prediction. Experiments on small/moderate datasets show competitive PI quality against NNs and RFs with far fewer parameters.

Weaknesses: The biggest issue for me is the missing baselines. The paper frames the comparison as SVM vs NN vs RF, but in practice, the go-to for tabular UQ right now is gradient boosted trees like XGBoost or LightGBM with quantile loss, or NGBoost. These are fast, handle moderate-scale data well, and are what most practitioners would reach for. Not including them makes it hard to judge where SVMs actually stand in the current landscape.

Similarly, for the time-series experiments, the datasets are quite small (365, 464, 3651 points). Pretrained forecasting models like Chronos, TimesFM, or Moirai can handle these out of the box with no training at all, which kind of sidesteps the whole model uncertainty discussion. If you don't train, there's no run-to-run variance by definition. It would strengthen the paper to show whether SVMs still offer something over these zero-shot approaches.

**Audience:**

Yes

**Audience Explanation:**

UQ, conformal prediction, and kernel-method researchers in the TMLR audience would find the stability argument and the sparse-SVM feature-selection idea worth reading

**Broader Impact Concerns:**

None. The paper actually argues for more stable/reproducible UQ in data-limited settings like clinical applications, which is a reasonable positive framing

**Claims And Evidence:**

Yes

**Claims Explanation:**

Partially. SVMs are a preferable choice for UQ "in the current landscape" is not adequately supported, because the strongest modern competitors like GBDT-quantile/NGBoost for tabular are absent

**Requested Changes:**

Add gbdt-with-quantile-loss (xgboost or lightgbm) and ideally ngboost as baselines across the PI and conformal experiments. Report stability for these too, since fixed-seed gbdt are also near-deterministic.

Compare against at least one zero-shot foundation forecaster (chronos/timesfm/moirai) on the time-series datasets, or explicitly scope the claims to the trained-model setting and explain why the comparison is still meaningful.

---

> ### Author Response · Authors · 2026-07-13
>
> We gratefully acknowledge the reviewers’ thoughtful and constructive comments and believe that careful consideration of these suggestions has meaningfully strengthened the quality and clarity of our work. We have attached a response letter as supplementary material in our submission that presents our point-by-point responses to the reviewers’ observations and recommendations. A copy of the revised manuscript is attached with this response letter, where all changes have been highlighted in blue for ease of reference.
> We have significantly revised our manuscript in the light of reviewer's valuable comments and incorporated reviewer suggestions in the revised manuscript. Please see the response letter in supplementary material. We shall be happy to answer any future concerns or queries.

---

### Comment · Reviewer_MoP7 · 2026-06-03
**preliminary comment**

The paper argues that SVMs are underappreciated for uncertainty quantification in regression, mainly because they produce deterministic solutions (no run-to-run variance), can yield sparse quantile estimates via L1 regularization, and extend naturally to conformal prediction. Experiments on small/moderate datasets show competitive PI quality against NNs and RFs with far fewer parameters.

Weaknesses:
The biggest issue for me is the missing baselines. The paper frames the comparison as SVM vs NN vs RF, but in practice, the go-to for tabular UQ right now is gradient boosted trees like XGBoost or LightGBM with quantile loss, or NGBoost. These are fast, handle moderate-scale data well, and are what most practitioners would reach for. Not including them makes it hard to judge where SVMs actually stand in the current landscape.

Similarly, for the time-series experiments, the datasets are quite small (365, 464, 3651 points). Pretrained forecasting models like Chronos, TimesFM, or Moirai can handle these out of the box with no training at all, which kind of sidesteps the whole model uncertainty discussion. If you don't train, there's no run-to-run variance by definition. It would strengthen the paper to show whether SVMs still offer something over these zero-shot approaches.

---

### Author Response · Authors · 2026-07-02
**Regarding the extension of the deadline for the response and revised manuscript submission**

The authors express their sincere thanks to the reviewers and the Editor for completing the review of the submitted manuscript. The reviews are indeed very valuable, and we are working diligently to incorporate the reviewers' suggestions to further strengthen our manuscript. However, this process is taking more time than anticipated.

We kindly request the Editor to extend the deadline for the submission of the revised manuscript and the detailed response to the reviewers by 14 days. This additional time will help us improve the quality of our manuscript in light of the reviewers' suggestions.